# Comparison of paleobotanical and biomarker records of mountain peatland and forest ecosystem dynamics over the last 2600 years in Central Germany

Carrie L. Thomas[1,2], Boris Jansen[2], Sambor Czerwiński[3,4,5], Mariusz Gałka[6], Klaus-Holger Knorr[7], E. Emiel van Loon[2], Markus Egli[1], and Guido L. B. Wiesenberg[1]

[1]Department of Geography, University of Zurich, 8057 Zurich, Switzerland
[2]Institute for Biodiversity and Ecosystem Dynamics, University of Amsterdam, Amsterdam, 1098XH, The Netherlands
[3]Climate Change Ecology Research Unit, Faculty of Geographical and Geological Sciences, Adam Mickiewicz University, 61-680 Poznań, Poland
[4]Physical Geography Institute of Geography and Geology, University of Greifswald, 17489 Greifswald, Germany
[5]Department of Geomorphology and Quaternary Geology, University of Gdańsk, 80-309 Gdańsk, Poland
[6]Faculty of Biology and Environmental Protection, Department of Biogeography, Paleoecology and Nature Conservation, University of Lodz, 90-237 Łódź, Poland
[7]Institute for Landscape Ecology, Ecohydrology and Biogeochemistry, University of Münster, 48149 Münster, Germany

**Correspondence:** Carrie L. Thomas (carrie.thomas@geo.uzh.ch)

**Abstract.** As peatlands are a major terrestrial sink in the global carbon cycle, gaining understanding of their development and changes throughout time is essential to predict their future carbon budget and potentially mitigate the adverse outcomes of climate change. With this aim to understand peat development, many studies have investigated the paleoecological dynamics by analyzing various proxies, including pollen, macrofossil, elemental, and biomarker analyses. However, as each of these proxies is known to have its own benefits and limitations, examining them in parallel allows for a deeper understanding of these paleoecological dynamics at the peatland and a systematic comparison of the power of these individual proxies. In this study, we therefore analyzed peat cores from a peatland in Germany (Beerberg, Thuringia) to a) characterize the vegetation dynamics over the course of the peatland development during the late Holocene and b) evaluate to what extent the inclusion of multiple proxies, specifically pollen, plant macrofossils, and biomarkers, contributes to a deeper understanding of those dynamics and interaction among factors. We found that, despite a major shift in regional forest composition from primarily beech to spruce as well as many indicators of human impact in the region, the local plant population in the Beerberg area remained stable over time following the initial phase of peatland development up until the last couple of centuries. Therefore, little variation could be derived from the paleobotanical data alone. The combination of pollen and macrofossil analyses with the elemental and biomarker analyses enabled further understanding of the site development as these proxies added valuable additional information, including the occurrence of climatic variations, such as the Little Ice Age, and more recent disturbances, such as drainage.

# 1 Introduction

Peatlands have been well-established as an important sink in the terrestrial carbon cycle, containing about 25% (600 gigatons) of the global soil carbon stock (Yu et al., 2010), though only comprising 3% of global land area (Xu et al., 2018). The same characteristics that make peatlands an effective carbon sink, e.g., slow degradation rates and high accumulation rates of organic matter, also make them an excellent archive for paleoecological records at high temporal resolution (Barber et al., 1994; Chambers et al., 2012). Compared to other environmental archives, the use of peat sequences, particularly of ombrotrophic bogs, for investigating paleoenvironmental changes is advantageous as they are generally more accessible, contain material readily dated using radiocarbon resulting in chronologies with lower uncertainties, and are primarily influenced by atmospheric inputs only, thereby recording climatic information along with their ecologic response (Chambers et al., 2012). Furthermore, peat sequences provide some of the few well-resolved records of paleoenvironmental changes during the Holocene in central Europe, aside from lake sediments (e.g., Schwark et al., 2002; Hepp et al., 2019).

Through extensive studies across ecosystems, including peatlands, multiple proxies have been developed and applied to characterize paleovegetation and paleoclimatic conditions, including pollen (e.g., Speranza et al., 2000), macrofossils (e.g., Tuittila et al., 2007), and biomarkers (e.g., Ficken et al., 1998a; Xie et al., 2004). Although these proxies have all been reliably used in paleoenvironmental reconstructions, each has benefits and drawbacks in their application.

Pollen and plant macrofossils are commonly used paleobotanical analyses to a) reconstruct local and regional plant succession of peatlands and their surroundings, b) assess human or climate impact on plant communities and understand their response or resilience, and c) define pristine plant populations as reference conditions in their restoration (Speranza et al., 2000; Gałka et al., 2022b). Pollen records extracted from peatlands have a well-established history of use for reconstructing paleovegetation, beginning with Von Post's study of arboreal pollen in Swedish bogs (1916). Despite the benefits of using pollen to reconstruct vegetation, there can be several limitations. Pollen can be transported over large distances, meaning that the local vegetation signal is overlain by that of the region (Farrimond and Flanagan, 1996). Additionally, some plants produce more pollen than others, and some pollen is better-preserved, leading to over-representation of certain taxa in the results (Birks and Birks, 2000). These drawbacks can be mitigated by including macrofossil analysis alongside pollen (Birks and Birks, 2000). Due to waterlogged conditions and low pH, the remains of mosses, graminoids, and dwarf shrubs are well-preserved in peatlands, allowing for the identification of macrofossils to reconstruct the development and succession of in situ, peat-forming vegetation within peatlands (Chambers et al., 2012). Macrofossils often provide better taxonomic precision than pollen (Birks and Birks, 2000), which is important when reconstructing the potential of individual plants to aid in carbon sequestration (Sim et al., 2021). However, macrofossils can also be difficult to identify in highly humidified peat (Naafs et al., 2019). In such cases, or if changes are more subtle, plant biomarkers can be a valuable addition to studies of peat cores.

Plant biomarkers derived from leaf waxes have been used extensively in paleoecological investigations with a long history of research throughout the past century (e.g., Chibnall et al., 1934). The most commonly used of these markers are the straight-chain lipids: *n*-alkanes, *n*-alkanols, and *n*-fatty acids (Jansen and Wiesenberg, 2017). Previous studies of peatlands have used these biomarkers as proxies for paleoclimate and vegetation (e.g., Ficken et al., 1998a; Xie et al., 2004). As peat consists almost

entirely of organic matter, there are large concentrations of a range of biomarkers. While biomarkers may also be deposited as aerosols to a small extent, the vast majority derive directly from the parent peat vegetation, and even as the peat becomes more humified, biomarkers have the potential to persist (Naafs et al., 2019). Aliphatic compounds, including *n*-alkanes, are particularly resistant to decomposition and become residually enriched (Biester et al., 2014). In peatlands particularly, *n*-alkanes have been used to differentiate between vegetation types, as *Sphagnum* mosses typically produce shorter carbon chains (23-25), while longer carbon chains (27-33) are commonly produced by vascular plants (Chambers et al., 2012; Pancost et al., 2002; Baas et al., 2000). Indices comparing relative abundances of biomarker homologues, including the Carbon Preference Index (CPI), can also be used to indicate degradation of organic matter as well as climate changes that are more favorable for high microbial activity and enhanced degradation (Chambers et al., 2012). Yet, the use of biomarkers and biomarker-derived indices and ratios to reconstruct vegetation and other environmental conditions is complicated by the fact that there are multiple sources of biomarkers, and species-specific compositions are rare (Jansen and Wiesenberg, 2017). It is more straightforward to differentiate between vegetation types, but even this can be complicated due to factors such as moisture, humidity, and stage of plant/leaf development influencing biomarker distribution patterns (Jansen and Wiesenberg, 2017).

Previous studies have successfully combined pollen and lipid biomarker analyses to reconstruct regional paleoenvironmental conditions (e.g., Farrimond and Flanagan, 1996; Schwark et al., 2002), including in peatlands (e.g., Zhou et al., 2005). Additionally, a growing number of studies have included both pollen and macrofossil analyses along with biomarker measurements for a more robust interpretation of peat archives (e.g., Ronkainen et al., 2015; Balascio et al., 2020). Considering elemental analyses of carbon, nitrogen, and their stable isotopes also allows for more nuanced interpretations as the C/N ratio can be used as an indicator for decomposition (Kuhry and Vitt, 1996; Biester et al., 2014), as can $\delta^{13}$C for decomposition and hydrological changes (Loisel et al., 2010; Biester et al., 2014).

In this study, we aimed to evaluate these proxies individually and in combination while characterizing the paleovegetation dynamics of the Beerberg peatland, an ombrotrophic mountain peat bog in the Thuringian Forest (Germany). The Beerberg site was chosen as there are few paleoenvironmental archives of the late Holocene in Central Germany and, more specifically in the Thuringian Forest (e.g., Breitenbach et al., 2019), leading to a demand for better understanding of such scarce paleoenvironmental records (Githumbi et al., 2022). Local reconstructions of Thuringia are especially valuable as the area lies near the current boundary between the maritime Cfb and continental Dfb climatic zones in the Köppen-Geiger classification (Peel et al., 2007). This boundary shifts spatially through time and increased dated and high-resolution reconstructions are needed to finetune knowledge of the past evolution of the climate boundaries in central Europe (Breitenbach et al., 2019). Previous pollen analyses have been completed at the site by Jahn (1930) and Lange (1967) but were more limited in scope as Jahn only considered arboreal pollen and neither study included radiocarbon dating so there is no reliable chronology for the Beerberg peatland. Based on the previous, the aim of the present study was to assess a) the paleovegetation dynamics over time in the Thuringian forest over the last ca. 2600 years, an area currently lacking such continuous records, and b) how adding *n*-alkanols and *n*-fatty acids to a multi-proxy analysis helps to obtain a more detailed reconstruction of past environmental conditions and peatland development.

## 2   Materials and Methods

### 2.1   Study area and sampling

The Beerberg peatland (50° 39' 32" N, 10° 44' 36" E, 983 m) is a raised ombrotrophic peat bog at the summit of the Großer Beerberg in the Thuringian Forest (Germany) and is part of the Vessertal-Thuringian Forest Biosphere Reserve (Figure 1). The underlying bedrock is rhyolite (Lützner et al., 2012). Annual precipitation is estimated to be 1300 mm (Görner et al., 1984), and the mean annual temperature is 4°C (Jeschke and Paulson, 2000). While the Beerberg peatland has been under nature protection status since 1939, previously some of the peat was used as fuel for glass manufacturing. Consequently, the peatland was partially drained and reforested with spruce trees in the 19th and 20th centuries. Later, in particular after 1990, drainage trenches were filled and spruce trees removed to prevent further drying of the peat (Thüringer Landesanstalt für Umwelt und Geologie, 2002). Current vegetation includes *Sphagnum* mosses, such as *S. fuscum*, *S. magellanicum (s. l.)*, *S. angustifolium*, and *S. capillifolium*, tree species *Picea abies*, *Pinus sylvestris*, *Betula pendula*, and *Betula pubescens*, and other plants common in partially disturbed bogs, such as *Calluna vulgaris*, *Eriophorum vaginatum*, and *Polytrichum strictum*.

Sampling was completed in October 2019. Two Russian peat corers (5 cm diameter, Eijkelkamp, Giesbeck, The Netherlands; 7 cm diameter, self-made) were used to alternately core two small hummocks within approximately 20 cm distance to a depth of 340 cm. Overlapping core sections were taken from 90–100 cm, 175–190 cm, and 290–320 cm. The overlapping sections were correlated by sampling depth and stratigraphy, and for the elemental and biomarker analyses, the overlapping samples were treated as replicates, and the averages of the results were calculated.

### 2.2   Elemental analysis

For elemental analysis, samples were taken at 5 cm intervals, with a few exceptions due to visibly distinct layers in the peat at 10–12 cm, 170–172 cm, 270–272 cm, 325–327 cm, 327–328 cm, and 337.5–340 cm. The samples were freeze-dried to a constant weight and subsequently milled to a fine powder using a horizontal ball mill (MM400, Retsch). Approximately 1 mg of material per sample was used for the elemental analysis. Carbon and nitrogen concentrations (C, N), as well as stable C isotope values ($\delta^{13}$C), were measured using an Elemental Analyzer-Isotope Ratio Mass Spectrometer (EA-IRMS; FLASH 2000-HT Plus, linked by ConFlo IV to DELTA V Plus IRMS; Thermo Fisher Scientific). Calibration was carried out using caffeine (Merck, Germany) and a soil reference material from a Chernozem (Harsum, Germany; see Black Carbon Reference Materials, https://www.geo.uzh.ch/en/units/2b/Services/BC-material/Environmental-matrices.html). At least two analytical replicates were measured for all samples.

### 2.3   Radiocarbon dating

Hand-picked plant remains (Table A1) were selected from 12 depths (7.5 cm, 16.5 cm, 34.5 cm, 54.5 cm, 69.5 cm, 124.5 cm, 174.5 cm, 258.5 cm, 278.5 cm, 293.5 cm, 314.5 cm, and 334-336 cm). These were cleaned by an acid-alkali-acid treatment and combusted at 900°C to produce $CO_2$, which was reduced to graphite. The carbon isotope composition was measured

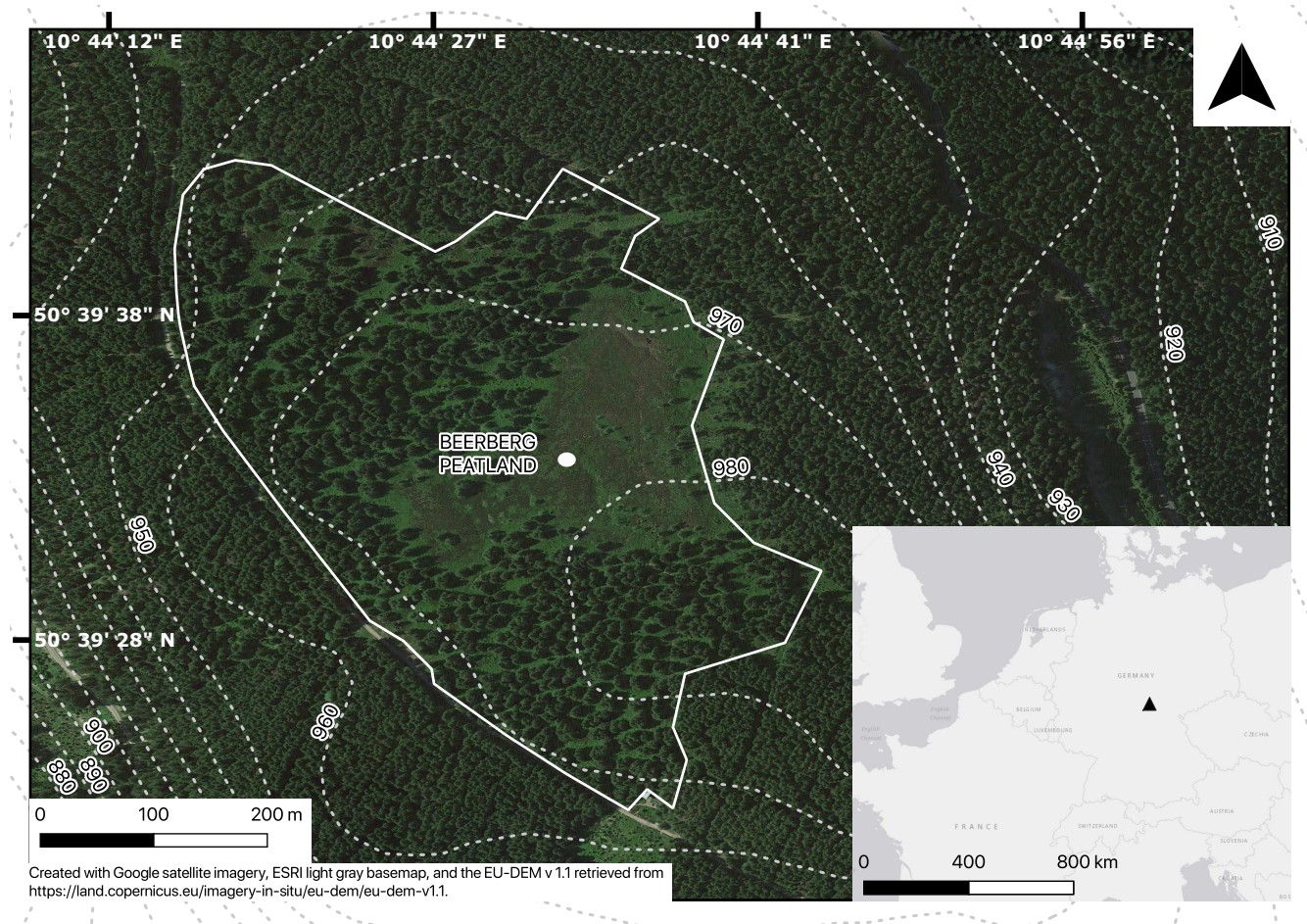

**Figure 1.** Map showing sampling spot in Beerberg peatland (area within continuous white boundary) with contour lines and inset showing location in continental Europe.

by Accelerator Mass Spectrometry (AMS) at the Institute of Ion Beam Physics at the Swiss Federal Institute of Technology (Zurich, Switzerland) using the 0.2 MV MICADAS facility.

### 2.4 Macrofossil analysis

Plant macrofossils were analyzed at 4 cm resolution (n=70, 1 cm thick peat slices) using samples of approximately 8 cm$^3$
volume. The samples were washed and sieved under a warm water current over 0.20 mm mesh screens. The percentage of individual fossils of vascular plants and brown mosses was estimated, and the fossil carpological remains and vegetative fragments (leaves, rootlets, epidermis) were identified using identification keys (Smith, 2004; Mauquoy and Van Geel, 2007) and compared to recently collected specimens. *S. capillifolium* and *Sphagnum rubellum* were grouped together (as *S. capillifolium/rubellum*) due to the difficulty of differentiating them in fossil state. Both species are typical ombrotrophic mosses that

occur together in relatively dry hummocks or lawns (Hölzer, 2010), with *S. rubellum* preferring moister habitats (Hölzer, 2010). Similarly, *Sphagnum medium* and *Sphagnum divinum* (in the past assigned to *S. magellanicum*, cf. Laine et al. (2018)) are expressed as *S. medium/divinum* because it was impossible to distinguish them based on morphological features in fossil state.

## 2.5 Pollen analysis

For pollen analysis, the core was sampled at 2.5 cm, 6.5 cm, 9.5 cm, and then 5 cm intervals starting at a depth of 12.5 cm, ending with a total of 69 samples. At each depth, a volume of 2 $cm^3$ was extracted and subjected to standard laboratory procedures for pollen analyses (Berglund and Ralska-Jasiewiczowa, 1986). Samples were treated with 10% hydrochloric acid (HCl) to dissolve carbonates, heated in 10% potassium hydroxide (KOH) to remove humic compounds, and finally soaked in 40% hydrofluoric acid (HF) for at least 24 h to remove the mineral fraction. One Lycopodium tablet (10679 spores; produced

by Lund University) was added to the samples (Stockmarr, 1971). Sample slides were analyzed using an ECLIPSE 50i upright microscope and counted to a sum of at least 500 arboreal pollen (AP) grains. However, due to the high peat accumulation rate, this sum was not achieved in 33 samples, including seven samples in which a 100 AP sum was not achieved. Pollen taxa were identified using atlases (Beug, 1961; Moore et al., 1991) and the reference grains owned by the Institute of Geoecology and Geoinformation, Adam Mickiewicz University, Poznań. Selected non-pollen palynomorphs (NPPs), such as fungi and algae,

and microscopic charcoal particles (size fractions: 0.01–0.1 mm; >0.1 mm) were also counted. Microscopic charcoal particles were counted until their number summed with simultaneously counted *Lycopodium* spores reached 200 (Finsinger and Tinner, 2005; Tinner and Hu, 2003). Palynological indicators of human impact were organized according to Behre (1981) and Gaillard (2013).

## 2.6 Biomarker analysis

For biomarker analysis, the same samples were used as for the elemental analysis. Soxhlet extraction was performed using 468 mg to 758 mg of the milled peat samples, as described by Wiesenberg and Gocke (2017). Briefly, total lipids were extracted over approximately 30 hours with dichloromethane (DCM): methanol (MeOH) (93:7, v/v). These extracts were then separated sequentially into three fractions containing, respectively, neutral components including *n*-alkanes and *n*-alkanols, *n*-fatty acids, and polar and high molecular weight compounds. For the separation, a glass column containing Silica 60 + 5% potassium

hydroxide (KOH), 63–200$\mu$m, was used along with the solvents DCM, DCM:formic acid (99:1, v/v), and DCM:MeOH (1:1, v/v). The neutral fraction was further separated into aliphatic, aromatic, and heterocompound fractions. For this, a pasteur pipette containing activated silica gel (100Å, 70–230 mesh, dried for at least 8 hours at 110°C) was used along with the solvents *n*-hexane, *n*-hexane:DCM (1:1, v/v), and DCM:MeOH (93:7, v/v).

  Prior to measurement, an internal standard of deuterated *n*-eicosanoic acid ($D_{39}C_{20}$, Cambridge Isotope Laboratories, Inc.)

was added to the fatty acid fraction, which was then methylated using a boron trifluoride-methanol solution (CAS #373-57-9, Sigma-Aldrich). Additionally, an internal standard of deuterated *n*-octadecanol ($D_{37}C_{18}$, Cambridge Isotope Laboratories, Inc.) was added to the heterocompound fraction, containing *n*-alkanols, which were then silylated using N,O-bis(trimethylsilyl)-

acetamide (BSA) (CAS #10416-59-8, Sigma-Aldrich). An internal standard of deuterated tetracosane ($D_{50}C_{24}$, Cambridge Isotope Laboratories, Inc.) was added to the aliphatic fraction before analysis. The $n$-alkanes, $n$-alkanols, and $n$-fatty acids were quantified on a GC (Agilent 7890B) equipped with a multimode inlet and a flame ionization detector (FID). Compound identification was performed on an Agilent 6890N GC equipped with a split-splitless injector coupled to an Agilent 5973 mass selective detector (MS). Both instruments were equipped with a DB-5MS column (50 m × 0.2 mm x 0.33 $\mu$m) and 1.5 m de-activated pre-column, with helium as the carrier gas (1 ml min$^{-1}$). The GC oven temperature for $n$-alkanes was held at 70°C for 4 min and increased to 320°C at a rate of 5°C min$^{-1}$ held for 50 min. For $n$-fatty acids and $n$-alkanols, the temperature was held at 50°C for 4 min, then increased to 150°C at a rate of 4°C min$^{-1}$, and finally increased to 320°C at 3°C min$^{-1}$ held for 40 min. The samples (1 $\mu$l) were always injected in splitless mode. The GC-MS was operated in electron ionization mode at 70 eV and scanned from m/z 50–550. Individual compounds were identified by comparison of mass spectra with those of external standards and from the NIST and Wiley mass spectra library.

## 2.7  Data processing and analysis

### 2.7.1  Data entry, processing, and analysis

Data were entered, organized, and screened in Microsoft Excel. Subsequent data processing and analysis were conducted in R version 4.0.4 (R Core Team, 2021). The data were combined in five tables, one each for the radiocarbon dating, elemental analysis, plant macrofossils, pollen, and biomarker composition. In each table, each row represented one depth and each column one parameter. All data are available at Pangaea via https://doi.org/10.1594/PANGAEA.961142.

### 2.7.2  Elemental analysis

The mean and standard error were calculated for the analytical replicates used in the elemental analysis. To be consistent across all of the data interpretation and to determine the locations of potential climate and vegetation changes in the core, we applied a hierarchical clustering analysis most commonly performed on vegetation abundance data. We used the R packages vegan (Oksanen et al., 2020) and rioja (Juggins, 2020). The Euclidean distance was determined using the concentrations of C and N, the C/N ratio, and the C stable isotope values in the function *dist*. A constrained hierarchical clustering approach (CONISS, Grimm (1987)) was performed on the dissimilarity using the function *chclust*, with clusters constrained by depth. The number of zones was determined using the broken-stick model (MacArthur, 1957; Bennett, 1996) with the function *bstick*.

### 2.7.3  Radiocarbon dating

An age-depth model was developed using the Bacon model (Blaauw and Christen, 2011), as implemented in the package rbacon (Blaauw et al., 2021). The uppermost radiocarbon date was excluded from the age-depth model, as it was from a sample within 10 cm of the surface, and the yielded date of -552 yr BP $\pm$ 23 is too young and stratigraphically inconsistent with the rest of the core and represents a postbomb peak of atmospheric $^{14}$C, following nuclear weapons testing in the 1950s and 60s (Table A1). As a constraint, an estimated surface sample age was added of -70 yr BP $\pm$ 5. IntCal20 was used as the calibration curve

(Reimer et al., 2020), and NH1 was used as the calibration curve for postbomb dates (Hua et al., 2013). All of the dates referred
to in the following sections are the mean values returned by the age-depth model, which were calculated at 1 cm resolution.

### 2.7.4 Macrofossil

A CONISS cluster analysis (Grimm, 1987) was performed on the macrofossil abundance data. In contrast with the elemental
data, the Bray-Curtis dissimilarity was calculated as this is a better measure for purely compositional data. This was completed
using the functions *vegdist*, *chclust*, and *bstick*, as they are described in the previous section.

### 2.7.5 Pollen and microcharcoal

The microscopic charcoal accumulation rate (CHAR$_{micro}$; unit: particles cm$^{-2}$ year$^{-1}$) was calculated as follows:

$$CHAR_{micro} = CHAC_{micro} * AR_{deposits} \tag{1}$$

in which CHAC$_{micro}$ is the concentration of the microscopic charcoal particles (unit: particles cm$^{-3}$), and AR$_{deposits}$ is the peat
or sediment accumulation rate (unit: cm year$^{-1}$) (Davis and Deevey, 1964).

Pollen percentages were calculated as taxon percentages with

$$taxon\ percentages = (number\ of\ taxon\ grains/TPS) \times 100\% \tag{2}$$

where TPS indicates the total pollen sum, including the AP and non-arboreal pollen (NAP) taxa, and excluding the local taxa
(i.e., aquatic, wetland, and spore-producing).

TILIA software was used to plot the diagram presenting the results of the palynological analysis (Grimm, 1993). We per-
formed a CONISS cluster analysis as described in the previous section using the absolute pollen counts. Only taxa present
in at least three samples and that reached at least three percent relative abundance in one sample were included. Non-pollen
palynomorphs and coprophilous fungi were not included in the analysis.

### 2.7.6 Biomarker

Biomarker amounts are reported as absolute concentrations in $\mu$g/g. The total lipid extract (TLE) was calculated in mg/g.
The Carbon Preference Index (CPI) (Marzi et al., 1993) and Average Chain Length (ACL) (Poynter et al., 1989) were also
calculated for the *n*-alkanes using the following equations:

$$CPI = \frac{(\sum_{i=n}^{m} C_{2i+1} + \sum_{i=n+1}^{m+1} C_{2i+1})}{2(\sum_{i=n+1}^{m+1} C_{2i})} \tag{3}$$

$$ACL = \frac{\sum_{i=n}^{m} (2i+1) * C_{2i+1}}{\sum_{i=n}^{m} C_{2i+1}} \tag{4}$$

where C$_x$ is the concentration of each lipid containing *x* carbon atoms; *n* and *m* are the chain lengths of, respectively, the
starting and ending lipids divided by 2 (note: both 2*n* and 2*m* should be even numbers). For the *n*-alkanes, *m* is 11 and *n* is 15.

For the *n*-alkanols and *n*-fatty acids, the formulas were slightly adjusted to:

$$CPI_{ALC/FA} = \frac{(\sum_{i=n}^{m} C_{2i} + \sum_{i=n+1}^{m+1} C_{2i})}{2(\sum_{i=n+1}^{m+1} C_{2i+1})} \tag{5}$$

$$ACL_{ALC/FA} = \frac{\sum_{i=n}^{m} (2i) * C_{2i}}{\sum_{i=n}^{m} C_{2i}} \tag{6}$$

For the *n*-alkanols, *m* is 10 and *n* is 14. For the *n*-fatty acids, *m* is 10 and *n* is 16.

A number of indicative ratios have been developed to help interpret biomarker data, particularly for *n*-alkanes. We applied many of these to our data to see which was most applicable in the Beerberg setting. The $P_{aq}$ (Ficken et al., 2000) and $P_{wax}$ (Zheng et al., 2007) proxies have been previously used in sediments to differentiate between aquatic macrophyte and terrestrial plant input and in peatlands to infer past water levels with high $P_{aq}$ values associated with a higher water table and high $P_{wax}$ values associated with a lower water table (e.g., Zheng et al., 2007; Zhou et al., 2005; Nichols et al., 2006; Andersson et al., 2011).

$$P_{aq} = \frac{C_{23} + C_{25}}{C_{23} + C_{25} + C_{29} + C_{31}} \tag{7}$$

$$P_{wax} = \frac{C_{27} + C_{29} + C_{31}}{C_{23} + C_{25} + C_{27} + C_{29} + C_{31}} \tag{8}$$

Andersson et al. (2011) found that the $P_{aq}$ and $P_{wax}$ could be misleading if *S. fuscum* and *Betula* species were present in the peat due to their relatively high abundances of the $C_{23}$ homologue. They derived the ratio of $C_{23}/(C_{27}+C_{31})$ to correct for these inputs, and as both *S. fuscum* and *Betula* species were present at the Beerberg site, this ratio was included. The ratio of *n*-alkanes $C_{23}/C_{25}$ was also calculated, as this has been used in peatland settings before to determine shifts in *Sphagnum* species (McClymont et al., 2008). Further, $C_{23}/(C_{23}+C_{29})$ and $C_{25}/(C_{25}+C_{29})$ were calculated as well as other studies have used these to distinguish *Sphagnum* species in peat settings (e.g., Ronkainen et al., 2013). Far fewer ratios have been identified for *n*-alkanols and *n*-fatty acids. Nevertheless, we tested $C_{22}/C_{24}$, $(C_{22}+C_{24})/(C_{26}+C_{28})$ (Zheng et al., 2011), and $C_{24}/C_{28}$ for the *n*-alkanols and $C_{15}/(C_{15}+C_{16})$ (Zheng et al., 2007) and $C_{24}/C_{28}$ for the *n*-fatty acids.

A CONISS cluster analysis (Grimm, 1987) was performed on the absolute concentrations of the measured homologues of the *n*-alkanes ($C_{19}$–$C_{33}$), *n*-alkanols ($C_{16}$–$C_{28}$), and *n*-fatty acids ($C_{14}$–$C_{32}$) to determine whether different phases could be distinguished using only the biomarker data. The same procedure was followed as previously described in the macrofossil and pollen sections.

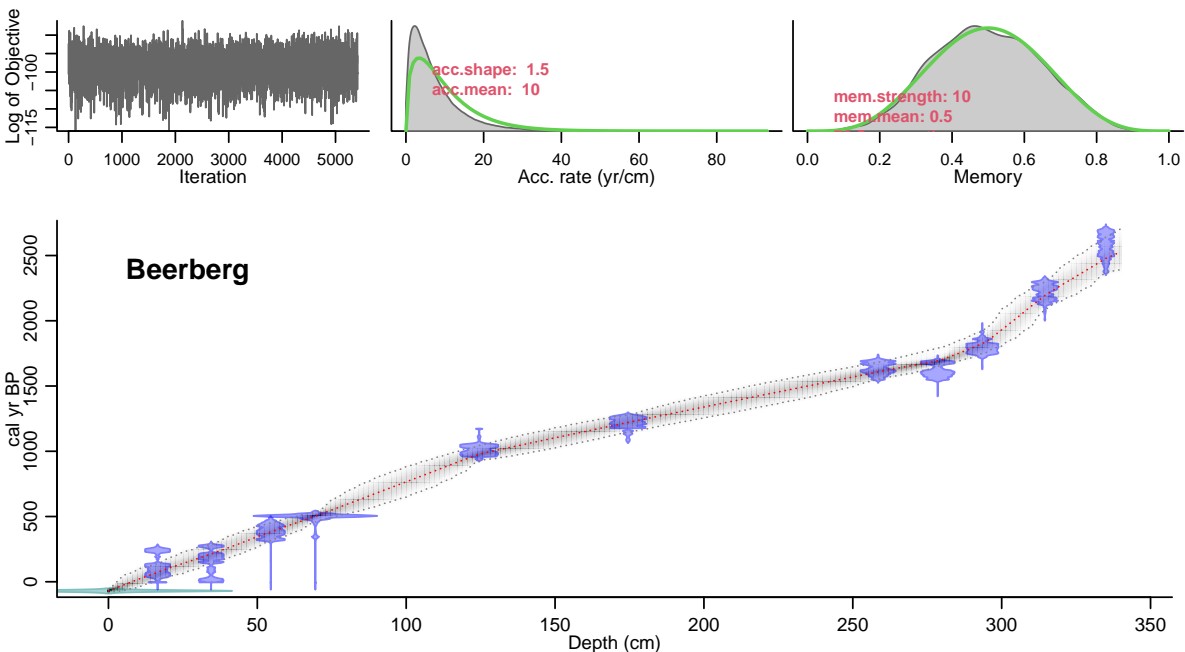

**Figure 2.** Age-depth model of the Beerberg core. For precise radiocarbon dates, see Table A1. The age-depth model is shown in the main graph. The distribution of the calibrated $^{14}$C dates is shown in blue and the estimated surface age of -70 cal yr BP is shown in green. The wider the distribution, the less precise the dates. The dashed, red curve shows the mean ages derived from the model and the dashed grey curves represent the 95% confidence intervals. The top three graphs in the figure show, respectively, the fit of the Markov Chain Monte Carlo (MCMC) iterations, the accumulation rate (yr/cm) with prior (green) and posterior (grey) distributions, and the prior (green) and posterior (grey) distributions for the memory, or how much the accumulation rate is able to change from one depth to the next. For more information on the Bacon model, see Blaauw and Christen (2011).

## 3   Results

### 3.1   Radiocarbon dating

The age-depth model that best matched the data in this study used a mean accumulation rate of 1 mm per year. The resulting
curve is visualized in Fig. 2. There are three clear phases with distinct accumulation rates in the model: 0.66 mm/yr from 340–293.5 cm (2528–1826 cal yr BP), 1.99 mm/yr from 293.5–124.5 cm (1826–978 cal yr BP), and 1.27 mm/yr from 124.5–0 cm (978–Present). Hence, the changes in accumulation rates correspond with the dated samples at 293.5 cm and 124.5 cm.

### 3.2   Elemental analysis

The results of the elemental analysis are shown in Fig. 3. The CONISS analysis based on Euclidean distance resulted in five
zones: 2528–2113 cal yr BP (Phase I-G), 2113–1569 cal yr BP (Phase II-G), 1569–1151 cal yr BP (Phase III-G), 1151–809 cal

yr BP (Phase IV-G), and 809 cal yr BP–Present (Phase V-G). The transitions between phases appear to be most correlated with changes in the $\delta^{13}$C values. Carbon concentration (C) ranged from 41.1–51.1%, with one exceptionally low value of 21.1% in the basal sample of Phase I-G (340–310 cm). The nitrogen concentration (N) also had a sharp peak of 2.5% in Phase I-G at 2368 cal yr BP (327 cm). Both the C/N ratio and $\delta^{13}$C values increased during Phase I-G to 64.3 and -25.8, respectively. This increasing trend continued for both into Phase II-G (310–250 cm) with the C/N ratio averaging 90.2 and $\delta^{13}$C averaging -25.4. N values decreased while C stayed stable. During Phase III-G (250–160 cm), the C/N ratio fluctuated rapidly between 90.1–197.7, likely due to relatively small shifts in N. This phase contained the highest C/N values with low N as well as less negative $\delta^{13}$C values, ranging from -23.5– -25.2. In Phase IV-G (160–105 cm), C/N and $\delta^{13}$C began to decrease, ranging from 95.1–159.8 and -24.1– -25.6, respectively. In Phase V-G (105–0 cm), the C/N ratio continued decreasing as N values increased, reaching a peak of 2% at 102 cal yr BP (20 cm). While the $\delta^{13}$C value of the topmost sample at -29.3‰ was particularly low, there was also an interval of increasing values from 593 cal yr BP (80 cm) to 345 cal yr BP (50 cm), reaching a local maximum of -24.5.

### 3.3 Macrofossil analysis

Based on the macrofossil analysis, the primary peat-forming species were *S. fuscum* and *S. medium/divinum* (Fig. 4). *S. fuscum* was dominant over most of the core, from 2086–106 cal yr BP (308.5–20.5 cm), with *S. medium/divinum* taking over in more recent layers (47– -48 cal yr BP; 13.5–3.5 cm). Additionally, *E. vaginatum* was an important species with two major periods from 64– -4 cal yr BP (15.5–7.5 cm) and 2528–2251 cal yr BP (340–318.5 cm).

The CONISS analysis based on the Bray-Curtis dissimilarity resulted in four significant phases (Fig. 4). Phase I-M (2528–2251 cal yr BP) is characterized by a high abundance of *E. vaginatum* as well as a relatively large amount of macrocharcoal (here charred wood pieces), especially at depth 328.5 cm (2388 cal yr BP). In Phase II-M (2251–1671 cal yr BP), *S. fuscum* was dominant. In Phase III-M (1671–64 cal yr BP), *S. fuscum* remained dominant, but there was also a small but relatively steady presence of *E. vaginatum* and Ericaceae rootlets. In Phase IV-M (64 cal yr BP–Present), *S. medium/divinum* replaced *S. fuscum*, and there was an increase in Ericaceae rootlets as well as *E. vaginatum*.

### 3.4 Pollen analysis

The CONISS cluster analysis of the complete pollen assemblage resulted in a separation into four phases, each representing a different regional vegetation composition (Fig. 5).

#### 3.4.1 Phase I-P (2528–1816 cal yr BP)

At the beginning of Phase I-P (340–292.5 cm), forests were dominated by *Fagus sylvatica* (pollen: 24–44.5%) (Fig. 5). *P. sylvestris*, *Betula* undiff., *Alnus* undiff., *Abies alba*, and *P. abies* also constituted a high proportion within the arboreal pollen. The latter two species significantly increased their percentage approaching 1810 cal yr BP. This time interval was characterized by high fire activity events as indicated by high CHAR$_{micro}$ (max 6035–38139 particles/cm$^2$/yr) values and the presence of

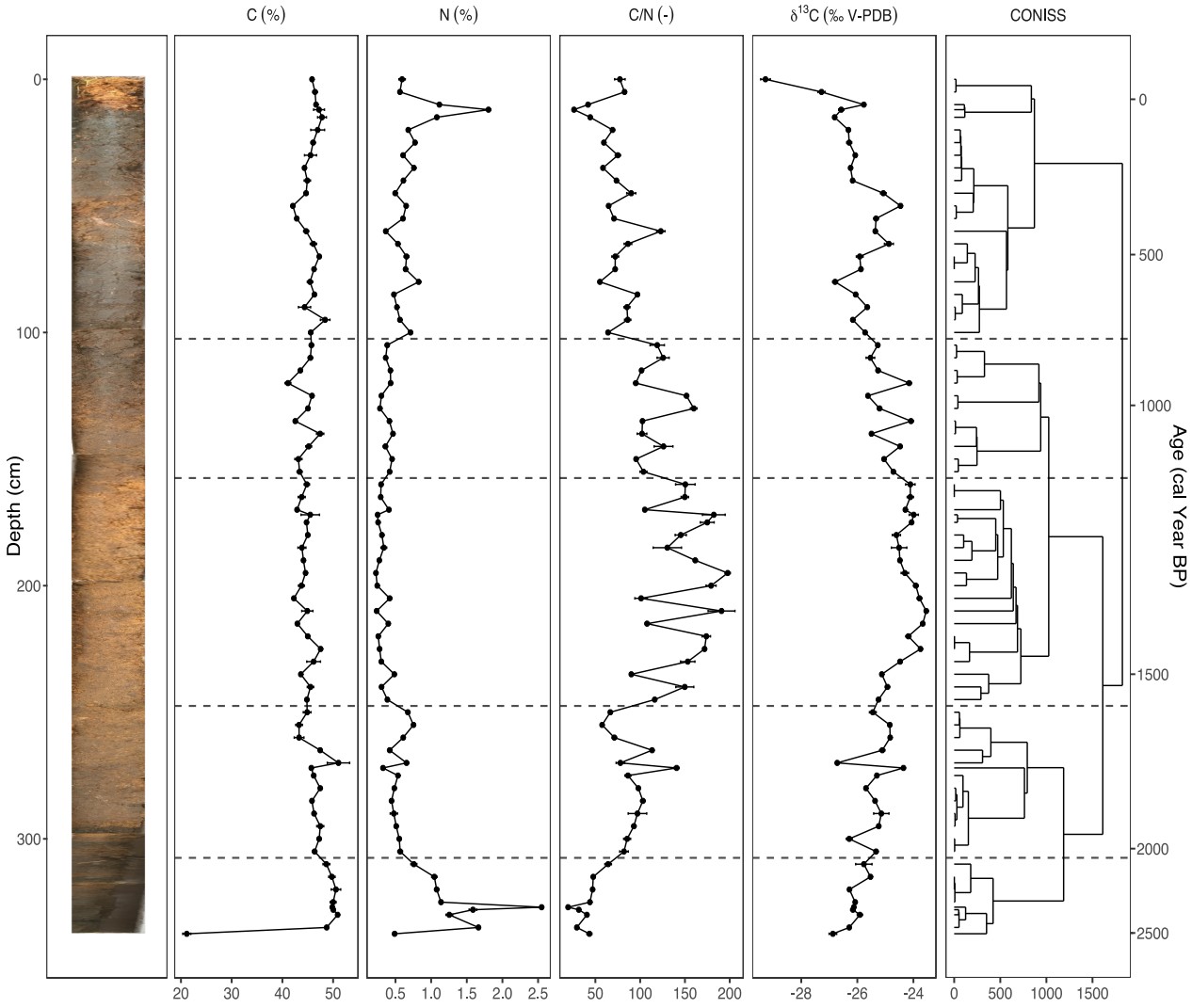

**Figure 3.** Carbon (C) and nitrogen (N) concentrations, C/N ratio, and stable carbon isotope composition ($\delta^{13}$C) plotted against depth (cm) and modeled age (cal year BP), including the resulting dendrogram of the CONISS analysis and phase boundaries indicated at 2113 cal yr BP (310 cm), 1569 cal yr BP (250 cm), 1151 cal yr BP (160 cm), and 809 cal yr BP (105 cm). An image of the sampled core is inset on the left. Two analytical replicates of each sample were measured. Error bars showing standard error are present for C and N concentrations, C/N ratio, and $\delta^{13}$C values.

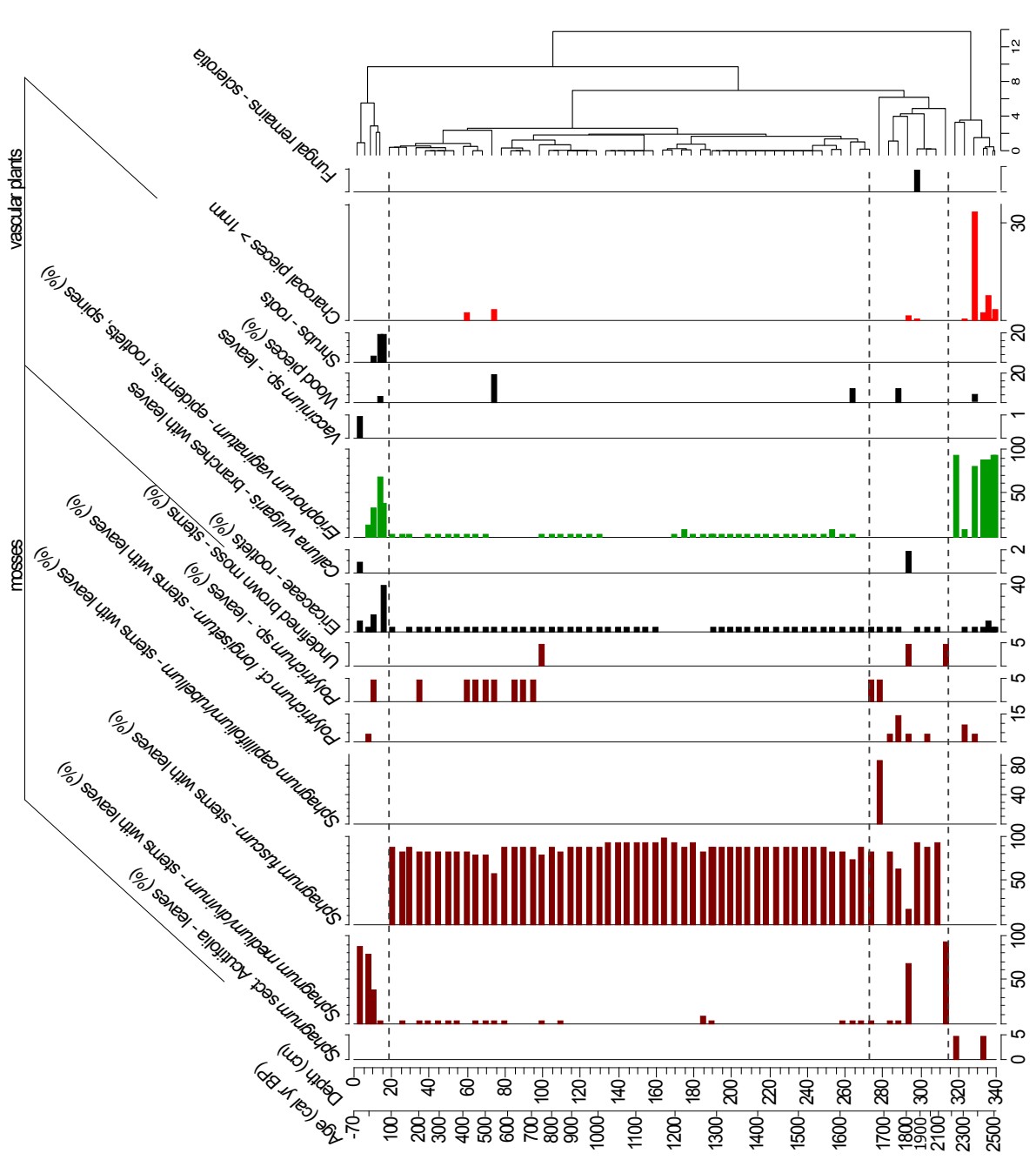

**Figure 4.** Plant macrofossil diagram presenting local vegetation development in the Beerberg peatland. Taxa with % are estimated volume percentages; the others are absolute counts (note scale differences on the x-axes). The resulting dendrogram of the CONISS analysis is on the right with phase boundaries indicated at 2251 cal yr BP (318.5 cm), 1671 cal yr BP (273.5 cm), and 64 cal yr BP (15.5 cm).

*Neurospora*, as well as *Gelasinospora* ascospores between 2500–2300 cal yr BP (Stivrins et al., 2019). Towards the end of the phase, *Sphagnum* began to increase, indicating a shift to moss-dominated peat.

### 3.4.2 Phase II-P (1816–1092 cal yr BP)

During the second phase, the forests were dominated by *F. sylvatica* (18–54.5%) (Fig. 5). *A. alba* and *P. abies* were the main components of coniferous forests. A significant decline in *F. sylvatica* occurred between 1280–1210 cal yr BP. During this phase, *A. alba* and *Quercus* reached their highest proportion in the forest. Based on indicator pollen counts, human impact in this phase was the weakest along the entire paleorecord. At the end of this phase, crop introduction in the region was observed, mirrored by an increase in Cerealia pollen share. The stable conditions prevailed at that time in the peatland, which was dominated by *Sphagnum* and *C. vulgaris*, and other Ericaceae species.

### 3.4.3 Phase III-P (1092–366 cal yr BP)

Throughout this zone, arboreal pollen declined from 97.5% to 77.5% between 1090–570 cal yr BP (Fig. 5). A substantial decline in late successional species such as *F. sylvatica* and *C. betulus* was observed, especially at the end of this phase. Along with the decrease in the proportion of these species, pioneer trees such as *P. sylvestris*, *Betula*, and *Corylus avellana* constituted an increasing proportion among the woodlands. During this phase, a constant share of cultivated indicators (mostly Cerealia undiff. and *Secale cereale*) was also recorded, especially from 740 cal yr BP. At the same time, a sharp increase of $CHAR_{micro}$, as well as coprophilous fungi taxa, was observed. This corresponded with a sharp decrease in the proportion of *Sphagnum* (740–570 cal yr BP), the values of which increased again at the end of this phase. At the same time, the disappearance of *K. deusta* was noticeable during this phase.

### 3.4.4 Phase IV-P (366 cal yr BP–Present)

The major deciduous trees that previously formed the stand gradually withdrew from the site, as manifested by decreasing percentages of *F. sylvatica* (from 18 to 4.4%), *Quercus*, and *Corylus avellana* (Fig. 5). *P. abies* and *P. sylvestris* reached their highest proportions in the forest (13.5–58% and 21–32.5%, respectively), whereas *A. alba* seemed to decline completely, as evidenced by the disappearance of its percentage share.

### 3.5 Biomarker analysis

The TLE following Soxhlet extraction ranged between 20–66 mg/g, with an average of 35 mg/g. *n*-Alkanes were measured with chain lengths from $C_{19}$ to $C_{33}$, *n*-alkanols from $C_{16}$ to $C_{28}$, and *n*-fatty acids from $C_{14}$ to $C_{32}$. The total abundance of *n*-alkanes ranged from 80–1112 $\mu$g/g, *n*-alkanols from 47–469 $\mu$g/g, and *n*-fatty acids from 260–4224 $\mu$g/g. Overall, the signatures are indicative of higher-plant source material (Eglinton and Hamilton, 1967). The CONISS cluster analysis identified four phases with boundaries at 35 cal yr BP (12 cm), 809 cal yr BP (105 cm), and 1657 cal yr BP (270 cm) (Fig. 6). These four phases are described in the following sub-sections.

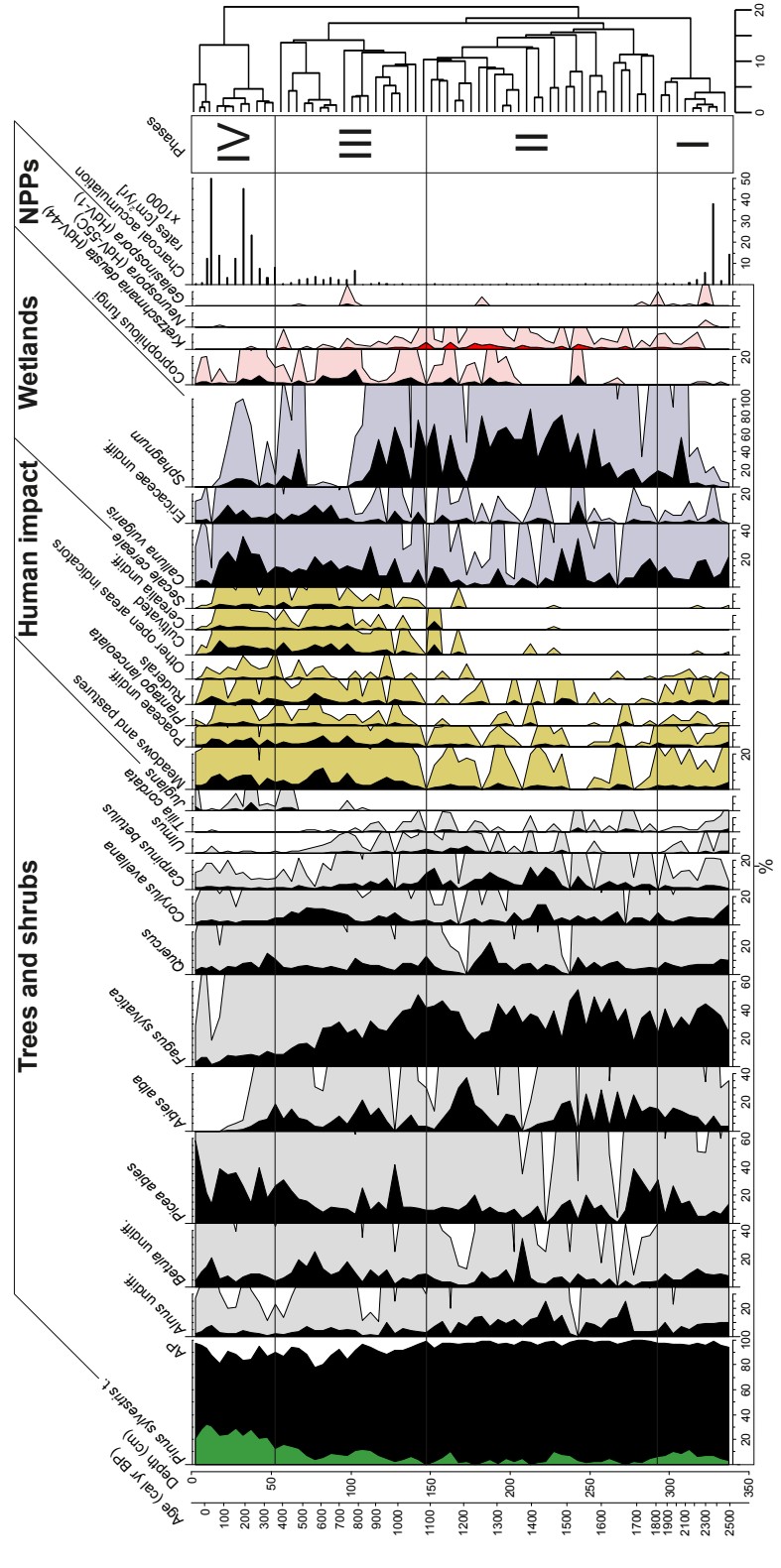

**Figure 5.** Diagram of selected pollen taxa, NPPs, and microscopic charcoal influx (CHAR$_{micro}$), including the resulting dendrogram of the CONISS analysis and phase boundaries indicated at 1816 cal yr BP (292.5 cm), 1092 cal yr BP (147.5 cm), 366 cal yr BP (52.5 cm).

### 3.5.1  Phase I-B (2528–1657 cal yr BP)

At the very beginning of the first phase, the TLE was relatively high but began to decrease around 2272 cal yr BP (320 cm). The most abundant homologues for the $n$-alkanes and $n$-fatty acids were consistently $C_{31}$ (33–508 $\mu$g/g) and $C_{24}$ (293–1376 $\mu$g/g), respectively (Fig. 6). For the $n$-alkanols, the most abundant varied between $C_{22}$, $C_{24}$, $C_{26}$, and $C_{28}$ (14–139 $\mu$g/g). The CPI of the $n$-alkanes ($CPI_{ALK}$) ranged from 6.7–14.6, averaging 12.3 (Fig. 7), that of the $n$-alkanols ($CPI_{ALC}$) from 8.7–12.8, averaging 10.7, and the CPI of the $n$-fatty acids ($CPI_{FA}$) from 3.9–8.2, averaging 6.7. The ACL of the $n$-alkanes ($ACL_{ALK}$) varied from 29.0–31.2, averaging 30.3, that of the $n$-alkanols ($ACL_{ALC}$) from 23.8–25.3, averaging 24.8, and that of the $n$-fatty acids ($ACL_{FA}$) from 24.4–25.1, averaging 24.9 (Fig. 7). Of the $n$-alkane ratios, $P_{aq}$ ranged from 0.05–0.33, averaging 0.13, $P_{wax}$ from 0.71–0.95, averaging 0.88, $C_{23}/(C_{27} + C_{31})$ from 0.02–0.18, averaging 0.07, $C_{23}/(C_{23}+C_{29})$ from .08–0.44, averaging 0.20, $C_{25}/(C_{25}+C_{29})$ from 0.13–0.59, averaging 0.27, and $C_{23}/C_{25}$ from 0.44–1.05, averaging 0.67. While $C_{23}/C_{25}$ and $P_{wax}$ generally decreased throughout the phase, $P_{aq}$, $C_{23}/(C_{23}+C_{29})$, $C_{25}/(C_{25}+C_{29})$, and $C_{23}/(C_{27}+C_{31})$ generally increased (Fig. 7). The $n$-alkanol ratios $C_{22}/C_{24}$ ranged from 0.55–1.00, averaging 0.75, $(C_{22}+C_{24})/(C_{26}+C_{28})$ from 0.62–2.03, averaging 1.03, $C_{24}/C_{28}$ from 0.34–0.70, averaging 0.52. Generally, $(C_{22}+C_{24})/(C_{26}+C_{28})$ and $C_{24}/C_{28}$ increased throughout the phase while $C_{22}/C_{24}$ decreased. The $n$-fatty acid ratios $C_{15}/(C_{15}+C_{16})$ ranged from 0.04–0.08, averaging 0.05, and $C_{24}/C_{28}$ from 0.72–0.90, averaging 0.81. $C_{15}/(C_{15}+C_{16})$ stayed about the same through the phase while $C_{24}/C_{28}$ increased.

### 3.5.2  Phase II-B (1657–809 cal yr BP)

In the second phase, the TLE decreased further until beginning to increase around 1209 cal yr BP (172 cm). The most abundant homologues for the $n$-alkanes and $n$-fatty acids remained consistently $C_{31}$ (19–317 $\mu$g/g) (with one exception of $C_{25}$ (26 $\mu$g/g) at 1223 cal yr BP (175 cm) and $C_{24}$ (123–747 $\mu$g/g), respectively (Fig. 6). For the $n$-alkanols, the most abundant varied between $C_{24}$, $C_{26}$, and $C_{28}$ (6–77 $\mu$g/g). All of the CPI values remained well over 1 (Fig. 7). The average of $ACL_{ALK}$ decreased slightly to 28.9, while the other ACL average values remained nearly the same. The averages of $P_{aq}$, $C_{23}/(C_{23}+C_{29})$, $C_{25}/(C_{25}+C_{29})$, and $C_{23}/(C_{27} + C_{31})$ increased to 0.35, 0.40, 0.55, and 0.19, while the averages of $P_{wax}$ and $C_{23}/C_{25}$ decreased to 0.70 and 0.53. The averages of the $n$-alkanol ratios $C_{22}/C_{24}$ and $(C_{22}+C_{24})/(C_{26}+C_{28})$ decreased to 0.49 and 0.92, while that of $C_{24}/C_{28}$ increased to 0.61. The averages of the $n$-fatty acid ratios stayed about the same. During this phase, many of the proxies followed a similar curve or its inverse, reaching either a peak or dip near the middle of the phase. These include $ACL_{ALK}$, $C_{23}/(C_{23}+C_{29})$, $C_{25}/(C_{25}+C_{29})$, $C_{23}/(C_{27} + C_{31})$, $P_{aq}$, $P_{wax}$, $n$-alkanol $C_{22}/C_{24}$, and $n$-fatty acid $C_{24}/C_{28}$ (Fig. 7).

### 3.5.3  Phase III-B (809–35 cal yr BP)

In the third phase, the TLE increased, though it had a slight dip from 466 cal yr BP to 259 cal yr BP (65–40 cm). Following the dip, the TLE increased again to the same level as the beginning of the phase. The most abundant homologues for the $n$-alkanes and $n$-fatty acids were consistently $C_{31}$ (55–423 $\mu$g/g) and $C_{24}$ (136–619 $\mu$g/g) (with one exception of $C_{26}$ (242 $\mu$g/g at 178 cal yr BP (30 cm)), respectively (Fig. 6). For the $n$-alkanols, the most abundant varied between $C_{24}$, $C_{26}$, and $C_{28}$ (3–103 $\mu$g/g). All of the CPI values again remained over 1 (Fig. 7). The averages of the ACL values all increased slightly (ALK: 30.3; ALC and

FA: 25.1). The averages of $P_{aq}$, $C_{23}/C_{25}$, $C_{23}/(C_{23}+C_{29})$, $C_{25}/(C_{25}+C_{29})$, and $C_{23}/(C_{27} + C_{31})$ decreased to 0.15, 0.52, 0.19, 0.31, and 0.07, while the average of $P_{wax}$ increased to 0.86. The average of the *n*-alkanol ratio $C_{22}/C_{24}$ increased to 0.62, while that of $(C_{22}+C_{24})/(C_{26}+C_{28})$ and $C_{24}/C_{28}$ decreased to 0.86 and 0.50. The average of the *n*-fatty acid ratio $C_{15}/(C_{15}+C_{16})$ increased due to a high peak at 250 cal yr BP (40 cm) then decreased to its previous level and stayed about the same, and that of $C_{24}/C_{28}$ decreased to 0.78. During this phase, many of the proxies reached a local maximum or minimum around 345 cal yr BP (50 cm) (Figs. 6 and 7).

### 3.5.4 Phase IV-B (35 cal yr BP–Present)

In the fourth, most recent phase, there are only four samples, so patterns are difficult to identify. The TLE initially dipped from its values in Phase III-B but was relatively high in the uppermost sample (43 mg/g). The most abundant homologues for the *n*-alkanes and *n*-fatty acids were consistently $C_{31}$ (42–75 $\mu$g/g) and $C_{24}$ (76–263 $\mu$g/g), respectively (Fig. 6). For the *n*-alkanols, the most abundant varied between $C_{22}$, $C_{24}$, $C_{26}$, and $C_{28}$ (8–43 $\mu$g/g). The CPI values all remained above 1 (Fig. 7). The average ACL values remained about the same. The average values of the *n*-alkane ratios stayed about the same except for $C_{23}/C_{25}$, which increased from 0.52 to 0.69 (Fig. 7). The averages of the *n*-alkanol ratios $C_{22}/C_{24}$ and $(C_{22}+C_{24})/(C_{26}+C_{28})$ increased to 0.76 and 0.96, while the averages of the other *n*-alkanol and *n*-fatty acid ratios stayed about the same.

## 4 Discussion

### 4.1 Temporal differences of trends across molecular proxies

The CONISS analysis identified different phases across all of the proxy records though there were similar trends across most of the records (Table 1). The elemental data split into five phases: 2528–2113 cal yr BP, 2113–1569 cal yr BP, 1569–1151 cal yr BP, 1151–809 cal yr BP, and 809 cal yr BP–Present (Fig. 3). The C and N concentrations, C/N ratio, and $\delta^{13}$C values are reflective of both the source material and environmental conditions, as well as post-deposition decomposition. Therefore, though there are similarities between the patterns of the elemental data and the other proxy records, they do not match exactly.

The macrofossil data was clustered into four phases: 2528–2251 cal yr BP, 2251–1671 cal yr BP, 1671–64 cal yr BP, and 64 cal yr BP–Present (Fig. 4). Compared to the elemental data, the CONISS analysis identified one fewer significant zones and while the first two phase boundaries of each proxy are within 140 cal yr BP of each other, the others do not align.

The pollen data was clustered into four phases: 2528–1816 cal yr BP, 1816–1092 cal yr BP, 1092–366 cal yr BP, and 366 cal yr BP–Present (Fig. 5). The pollen data reflects both a local and regional vegetation signal so while there are similarities between the phases based off of the pollen and those based off of the elemental and macrofossil data, they do not exactly line up.

The biomarker data was also clustered into four phases: 2528–1657 cal yr BP, 1657–809 cal yr BP, 809–35 cal yr BP, and 35 cal yr BP–Present (Fig. 6). The first three phases in the biomarker data are most similar to those of the pollen data, while

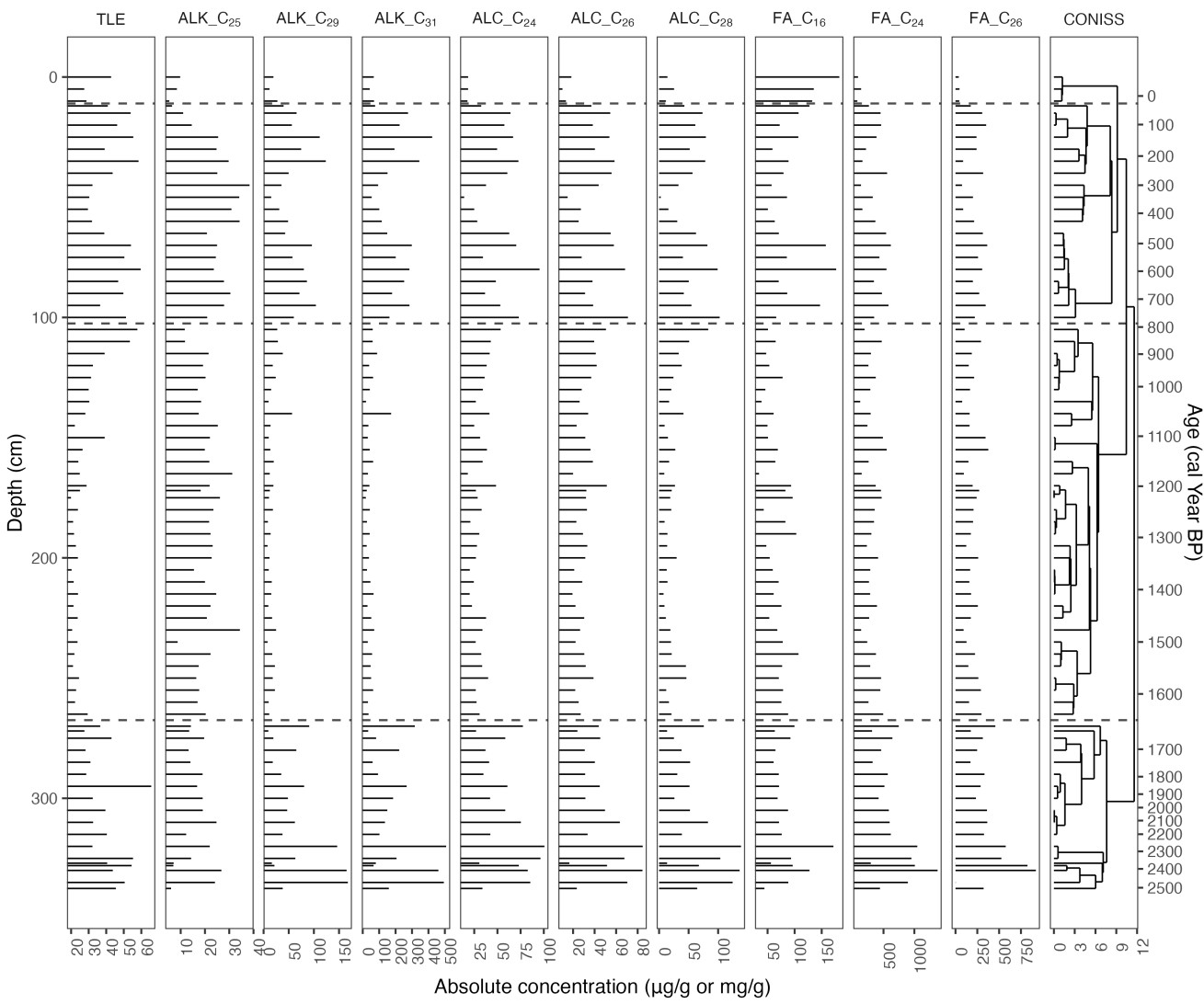

**Figure 6.** Absolute concentration (μg/g) values for most abundant homologues of the *n*-alkanes (ALK; C$_{29}$, C$_{31}$, and C$_{33}$), *n*-alkanols (ALC; C$_{24}$, C$_{26}$, and C$_{28}$), and *n*-fatty acids (FA; C$_{22}$, C$_{24}$, and C$_{26}$) as well as the C$_{23}$ and C$_{25}$ *n*-alkanes and C$_{16}$ *n*-fatty acid. On the right side is the resulting dendrogram of the CONISS analysis of the complete biomarker values with phase boundaries indicated at 35 cal yr BP (12 cm), 809 cal yr BP (105 cm), and 1657 cal yr BP (270 cm).

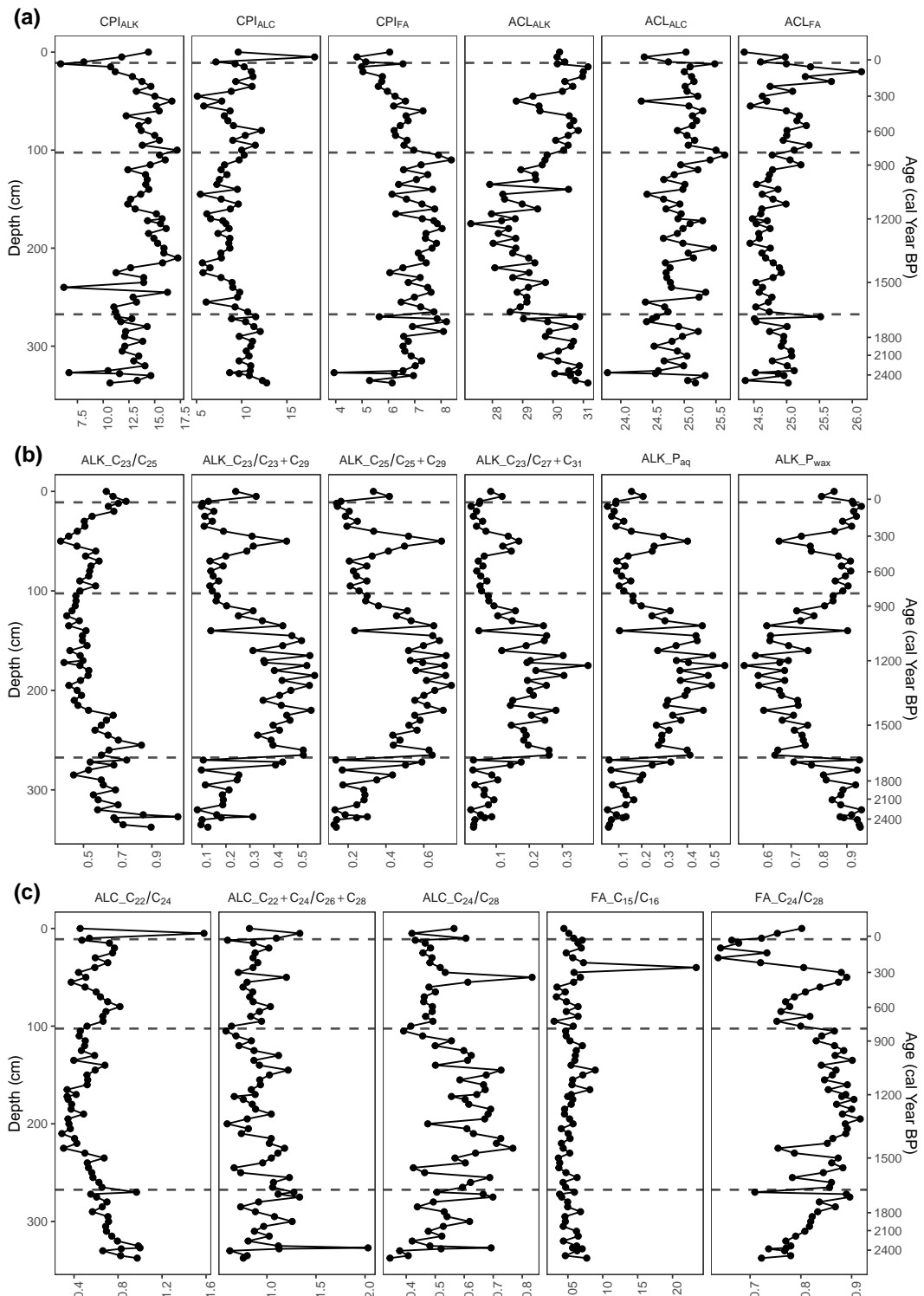

**Figure 7.** Carbon Preference Index (CPI) and Average Chain Length (ACL) values of *n*-alkanes (ALK), *n*-alkanols (ALC), and *n*-fatty acids (FA) and *n*-alkane, *n*-alkanol, and *n*-fatty acid ratios. The four phases from the cluster analysis are indicated at 35 cal yr BP (12 cm), 809 cal yr BP (105 cm), and 1657 cal yr BP (270 cm).

**Table 1.** Phases identified with CONISS analysis of different proxies and major changes between each phase.

| Depth (cm) | Age (cal yr BP) | Macrofossils (cal yr BP) | | Pollen (cal yr BP) | | Biomarkers (cal yr BP) | | C and N, $\delta^{13}$C (cal yr BP) | |
|---|---|---|---|---|---|---|---|---|---|
| 0 | Present | Phase IV-M 64—Present | ↓ *S. fuscum* | Phase IV-P 366—Present | ↓ *Sphagnum* | Phase IV-B 35—Present | ↑ FA C$_{16}$ | Phase V-G 809—Present | ↑ N |
|  | 0 |  |  |  | ↑ *C. vulgaris* |  | ↑ FA C$_{24}$/C$_{28}$ |  | ↓ C/N |
| 20 | 100 |  | ↑ *S. medium/divinum*<br>↑ Ericaceae<br>↑ *E. vaginatum* |  | ↑ Ericaceae |  | ↑ CPI$_{ALK}$ |  | ↓ $\delta^{13}$C |
|  | 200 | Phase III-M 1671—64 |  |  | ↑ *P. sylvestris* | Phase III-B 809—35 | ↑ TLE |  |  |
| 45 | 300 |  | ↓ *S. medium/divinum* |  | ↑ *P. abies* |  | ↑ P$_{wax}$ |  |  |
|  | 400 |  | ↑ *E. vaginatum* |  | ↓ *F. sylvatica*<br>↑ Charcoal |  | ↑ ALC C$_{22}$/C$_{24}$<br>↓ P$_{aq}$ |  |  |
| 70 | 500 |  | ↑ *Polytrichum* | Phase III-P 1092—366 | ↓ *Sphagnum* |  | ↓ FA C$_{24}$/C$_{28}$ |  |  |
|  | 600 |  |  |  | ↑ Ericaceae |  |  |  |  |
|  | 700 |  |  |  | ↑ *C. vulgaris* |  |  |  |  |
| 105 | 800 |  |  |  | ↑ Cultivated taxa |  |  |  |  |
|  | 900 |  |  |  | ↓ *F. sylvatica* | Phase II-B 1657—809 | ↓ TLE | Phase IV-G 1151—809 | ↓ C/N |
|  | 1000 |  |  |  |  |  | ↑ P$_{aq}$ |  | ↓ $\delta^{13}$C |
| 150 | 1100 |  |  |  |  |  | ↑ FA C$_{24}$/C$_{28}$ |  |  |
|  | 1200 |  |  | Phase II-P 1816—1092 | ↑ *Sphagnum* |  | ↓ P$_{wax}$ |  |  |
|  | 1300 |  |  |  | ↓ *P. abies* |  | ↓ ALC C$_{22}$/C$_{24}$ | Phase III-G 1569—1151 | ↑ C/N |
|  | 1400 |  |  |  | ↓ *P. sylvestris* |  |  |  | ↑ $\delta^{13}$C |
| 235 | 1500 |  |  |  | ↓ Charcoal |  |  |  |  |
|  | 1600 |  |  |  |  |  |  |  |  |
|  | 1700 |  |  |  |  |  |  | Phase II-G 2113—1569 | ↑ C/N |
| 290 | 1800 | Phase II-M 2251—1671 | ↑ *S. fuscum* | Phase I-P 2528—1816 | ↑ *Sphagnum* | Phase I-B 2528—1657 | ↓ TLE |  | ↑ $\delta^{13}$C |
|  | 1900 |  | ↑ *S. medium/divinum* |  | ↑ *P. abies* |  | ↑ P$_{aq}$ |  |  |
|  | 2000 |  | ↓ *E. vaginatum* |  |  |  | ↑ FA C$_{24}$/C$_{28}$ |  |  |
|  | 2100 |  | ↓ Charcoal |  |  |  | ↓ P$_{wax}$ |  |  |
| 315 | 2200 |  |  |  | ↓ Charcoal |  | ↓ ALC C$_{22}$/C$_{24}$ | Phase I-G 2528—2113 | ↑ C |
|  | 2300 |  |  |  |  |  |  |  | ↑ C/N |
|  | 2400 | Phase I-M 2528—2251 | ↑ Charcoal |  |  |  |  |  | ↑ $\delta^{13}$C |
| 340 | 2528 |  |  |  |  |  |  |  |  |

the final phase is most closely aligned with that of the macrofossil data. The biomarker data reflects the source material and post-deposition decomposition, so some offsets to the other proxy records are expected.

Because macrofossils are conventionally seen as the most reliable proxy reflecting in situ vegetation and peat development (Birks and Birks, 2000, e.g.,), we used the phases based on the macrofossils to describe the peatland development.

### 4.2   Development of the Beerberg peatland

In Phase I-M, 2528–2251 cal yr BP, the vegetation is characterized by a dominant presence of *E. vaginatum* (Fig. 4), along with a relatively low C/N ratio and very negative $\delta^{13}$C values (Fig. 3), which is typical for fen peat or transitional peat underlying bog peat (Kuhry et al., 1992; Jones et al., 2010). High fire activity during this phase was indicated by the maximum count of macrocharcoal pieces (charred wood, moss stems, *Calluna* leaves) (Fig. 4) occurring in this interval as well as increased CHAR$_{micro}$ (Fig. 5) and the presence of carbonicolus ascospores, *Neurospora* and *Gelasinospora* (Shumilovskikh and van Geel, 2020). It is likely that some of the fire events were a result of anthropogenic activity, indicated by the presence of *Plantago*

*lanceolata* or ruderal communities like *Artemisia* or Chenopodiaceae during this phase (Fig. 5). Fires in the bog area could have initiated the peat development as initial (fen) peat could form on wet ground following fire, developing into transitional peat, and then bog peat as has been seen at other sites (e.g., Tuittila et al., 2007; Gałka et al., 2019). The pollen analysis also indicated that *Fagus sylvatica* was the dominant arboreal species during this time period (Fig. 5), as was also determined by Lange (1967). Although *F. sylvatica* is considered a fire-sensitive species (Tinner et al., 2000), it records maximum occurrence

in this phase soon after the fire. This could also signify that the fire negatively affected the presence of *A. alba* and *P. abies* in contrast to *F. sylvatica*. The disappearance of *Neurospora* and *Gelasinospora* combined with the rapid decline in $CHAR_{micro}$ corresponds with the development of the *Sphagnum* community on peatland. This may also indicate wetter conditions on peatlands, especially during 2500-2230 cal yr BP. The increasing C/N ratio and $\delta^{13}C$ values towards the end of the phase (Fig. 3), as well as the initial incidences of *Sphagnum* (Figs. 4 and 5), indicates a shift towards more ombrotrophic conditions

(Wang et al., 2015). The estimated accumulation rate during this phase (Fig. 2) is also relatively low compared to the rest of the core, likely due to the lower proportion of mosses (Stivrins et al., 2017) as it is the paludification phase, the initiation of the peatland. The biomarker data also supports this interpretation with $P_{wax}$, $P_{aq}$, $C_{23}/(C_{23}+C_{29})$, $C_{25}/(C_{25}+C_{29})$, and $C_{23}/(C_{27}+C_{31})$ all indicating an initial high proportion of longer chain length *n*-alkanes usually deriving from vascular plants (Fig. 7). Through the phase, the relative abundance of the shorter $C_{23}$ and $C_{25}$ *n*-alkanes (Fig. 6) increases, indicating the beginning of

the peat development and an increase in the proportion of *Sphagnum* mosses (Baas et al., 2000; Pancost et al., 2002; Bingham et al., 2010). Of the other ratios, the *n*-alkanol $C_{22}/C_{24}$ generally decreased, while the *n*-fatty acid $C_{24}/C_{28}$ increased, potentially indicating an increase in *n*-alkanol $C_{24}$ and *n*-fatty acid $C_{24}$ which have been previously measured as dominant chain lengths in some *Sphagnum* species (e.g., Ficken et al., 1998a; Corrigan et al., 1976). Additionally, the TLE and individual biomarker absolute concentrations were relatively high compared to later phases in the core (Fig. 6). This aligns with previous research

finding that, compared to vascular plant species, *Sphagnum* mosses produce lower outputs of lipids (Pancost et al., 2002); therefore, as the proportion of *Sphagnum* increases, the TLE and biomarker concentrations decrease.

Following the shift to ombrotrophic conditions, in the second phase, 2251–1671 cal yr BP, *S. fuscum* was dominant and the main peat-forming plant (Fig. 4). There was also a fairly steady presence of Ericaceae rootlets, though no evidence of *E. vaginatum*. At the beginning of this phase, there were also remnants of *S. medium/divinum* species as well as *Polytrichum*.

In the pollen data, the percentage of *Sphagnum* spores rapidly increased, along with that of *P. abies* (Fig. 5). During this phase, the C/N ratio and $\delta^{13}C$ values began increasing (Fig. 3), indicating wetter conditions during rapid peat growth and low decomposition (Loisel et al., 2010; Kuhry and Vitt, 1996). The accumulation rate thus also began increasing during this phase (Fig. 2). In the biomarker data, the *n*-alkane ratios indicating an increase in the proportion of shorter chain lengths ($C_{23}$ and $C_{25}$) continued increasing due to the rise of *Sphagnum*. This was also the case for *n*-fatty acid $C_{24}/C_{28}$ (Fig. 7). The TLE

and absolute concentrations of individual biomarkers decreased from the first phase (Fig. 6), as expected with the increase in *Sphagnum*.

In the third phase, 1671–64 cal yr BP, *S. fuscum* was still dominant in the macrofossils, but a steady presence of *E. vaginatum* returned along with *S. medium/divinum* and *Polytrichum* in the later half of the phase (Fig. 4). The C/N ratio and $\delta^{13}C$ values reached their peak during this phase, indicating wet conditions and low decomposition (Loisel et al., 2010; Kuhry and Vitt,

1996). Within this interval, Phase II-P (1816–1092 cal yr BP) of the pollen assemblage is contained (Fig. 5). During this period, the presence of the parasitic fungus *Kretzschmaria deusta* - an indicator of tree fungal infections, often found on deciduous trees, especially on *Fagus sylvatica*, was recorded (Wilkins, 1934). Among other mountain sites, its presence was associated mostly with a higher proportion of *F. sylvatica* and other broad-leaved trees (Czerwiński et al., 2020; Kołaczek et al., 2020). The increased presence of this fungus in the past may have been related to stronger herbivore presence and/or coppicing prac-
tices and/or grazing damages (Latałowa et al., 2013; Karpińska-Kołaczek et al., 2014). However, the exact interactions between fungal infection and other disturbance factors in the past are not fully understood (Kołaczek et al., 2020). In the case of the Beerberg site, *K. deusta* was recorded during the period of lowest human impact and seemed to correspond with the develop­ment of forests with a higher role of *Carpinus betulus*. During this period, CHAR$_{micro}$ values were very low, which suggests a decline in fire activity or even a lack of fires near the study site. The C/N ratio and $\delta^{13}$C values also began to generally decline
(Fig. 3), which in an ombrotrophic peat indicates drier conditions (Loisel et al., 2010), increased decomposition (Kuhry and Vitt, 1996), and could be related to human impact, such as drainage of the peatland. In the forest, *P. abies* expanded (especially from 520 cal yr BP) and possibly replaced *F. sylvatica* in cleared areas (Fig. 5). This pattern has been observed previously at other sites in Germany, such as the Black Forest (Rösch, 2000; Gałka et al., 2022b). Deforestation was related to human im­pact, as evidenced by an increase in indicators typical for open landscapes, including meadows and pastures, such as Poaceae
and *Plantago lanceolata*, as well as ruderal habitats (mostly *Rumex acetosa/acetosella* type, *Artemisia*, Chenopodiaceae, and Brassicaceae undiff.). Further evidence of human impact results from the steady proportion of cultivated species, as well as the increase of CHAR$_{micro}$ and coprophilous fungi taxa. During this phase, the biomarker records showed more variation than the macrofossils. The TLE and absolute concentrations were relatively low through the phase until around 809 cal yr BP when they began to increase; there was another dip in the TLE and most individual concentrations (excluding *n*-alkanes C$_{23}$ and C$_{25}$)
from around 466 cal yr BP to 259 cal yr BP, increasing again in the final part of the phase (Fig. 6). The fluctuations in TLE are likely related to a decrease and subsequent increase in *Sphagnum*, as well as an increase in the proportion of Ericaceae species and other vascular plants (Pancost et al., 2002) in the latter half of the phase. The biomarker ratios follow a similar pattern, with many reaching either a maximum or minimum point around 1200 cal yr BP, including P$_{aq}$, C$_{23}$/(C$_{23}$+C$_{29}$), C$_{25}$/(C$_{25}$+C$_{29}$), and C$_{23}$/(C$_{27}$ + C$_{31}$), as well as the *n*-alkanol C$_{22}$/C$_{24}$ and *n*-fatty acid C$_{24}$/C$_{28}$ (Fig. 7). These ratios also showed a local maximum
or minimum around 345 cal yr BP. This is also seen in the $\delta^{13}$C curve as well as in a color change in the peat core itself (Fig. 3). All of these proxies appear to be indicating that there is first a decrease in mosses and water levels beginning around 809 cal yr BP, followed by an increase from around 500 cal yr BP to 345 cal yr BP, and then subsequent decrease. The earlier decrease around 809 cal yr BP, implying lower water table levels and drier conditions, could be related to the warmer Medieval Climate Anomaly (MCA; ca. 900–1400 CE) (Luterbacher et al., 2016). The following increase in water table levels or surface moisture
around 500 cal yr BP could be caused by colder, wetter conditions as a result of the Little Ice Age (LIA; ca. 1300–1850 CE), as similar changes during this interval have been noted in other peatland records from Germany (e.g., Barber et al., 2004) as well as Poland (Marcisz et al., 2020). Following this brief increase, the P$_{aq}$ decreased again while the P$_{wax}$ increased until around 35 cal yr BP (Fig. 7). This is likely linked to the increased human activity and drainage that occurred at the peatland in the 19th

and 20th centuries. As previously mentioned, peatlands throughout Europe have exhibited drier conditions in the most recent times (Swindles et al., 2019).

In the fourth phase, 64 cal yr BP–Present, *Sphagnum medium/divinum* was dominant, and there was an increased presence of *E. vaginatum* (Fig. 4). This could be indicative that the peatland is no longer pristine as the shift in dominant *Sphagnum* species could be related to pollution through increasing dust deposition (Gałka et al., 2019, 2022a, b) and further dry conditions, related to drainage as well as the warming climate (Swindles et al., 2019). However, in the pollen analysis, the decrease of human impact indicators following 35 cal yr BP could be evidence of the introduced conservation practices on or near the peatland (Fig. 5). This phase coincided with again more negative $\delta^{13}C$ (Fig. 3) due to the peat here being in early stages of decay (Loisel et al., 2010), and elevated N concentration, potentially caused by the increase in more fen-like vegetation including *E. vaginatum* (Kuhry et al., 1992). However, the high N concentration could also result from high atmospheric N deposition throughout the last decades (Ackerman et al., 2019). The biomarker signature in this phase seemed to primarily differ based on a higher abundance of the $C_{16}$ homologue of the *n*-fatty acids (Fig. 6). As this corresponds to the current acrotelm of the peatland with active peat formation and high decomposition, the higher abundance of microbial-derived biomarkers are a result of higher microbial activity (Ficken et al., 1998b).

## 4.3  Added value of biomarker analysis

Both the biomarker and macrofossil analyses are reflective of the local vegetation within the peatland, while the pollen analysis reflects both the local and regional vegetation. Timing differences between local and regional vegetation shifts have previously been reported in studies using biomarkers and pollen analyses (e.g., Jansen et al., 2013).

Despite potential discrepancies in the cluster analysis-derived phases, the biomarker results illustrate a similar story of peatland development. The primary driver behind the phase changes in the biomarker data appeared to be the abundance of $C_{25}$ *n*-alkane. As $C_{25}$, along with $C_{23}$, is known to be highly abundant in *Sphagnum* species, the phases resulting from the CONISS analysis most likely represent changes in the input of *Sphagnum* to the peat (Fig. 6). Comparing the biomarker results to those of the macrofossils and pollen, the increase in *Sphagnum* did generally correlate well (Figs. 4, 5, 6, 8). Moreover, the increase of *Sphagnum* and related biomarkers were paralleled by higher C/N ratios and enriched $\delta^{13}C$ values (Fig. 3), indicative of more ombrotrophic conditions (Wang et al., 2015). The *n*-alkanol ratio $C_{22}/C_{24}$ and the *n*-fatty acid ratio $C_{24}/C_{28}$ also roughly followed these trends. While they have not been previously linked to *Sphagnum* mosses as indicative ratios, future peat biomarker studies could investigate further.

One notable difference shown in the biomarker measurements as opposed to the macrofossils was the behavior of $P_{aq}$, $P_{wax}$, and $C_{23}/(C_{27} + C_{31})$, as well as *n*-alkanol ratio $C_{22}/C_{24}$ and the *n*-fatty acid ratio $C_{24}/C_{28}$ in Phase III-B (Fig. 7). The generally decreasing trend of $P_{aq}$ and $C_{23}/(C_{27}+C_{31})$ at the beginning of this phase, implying lower water table levels and drier conditions, which could be related to the warmer Medieval Climate Anomaly (MCA; ca. 900–1400 CE) (Luterbacher et al., 2016). Then from around 500 cal yr BP to 345 cal yr BP, $P_{aq}$ began to increase while $P_{wax}$ decreased. This shift in conditions is also reflected in the $\delta^{13}C$ values (Fig. 8) as well as an abrupt increase in *Sphagnum* in the pollen data but is not reflected in a notable change in the macrofossils.

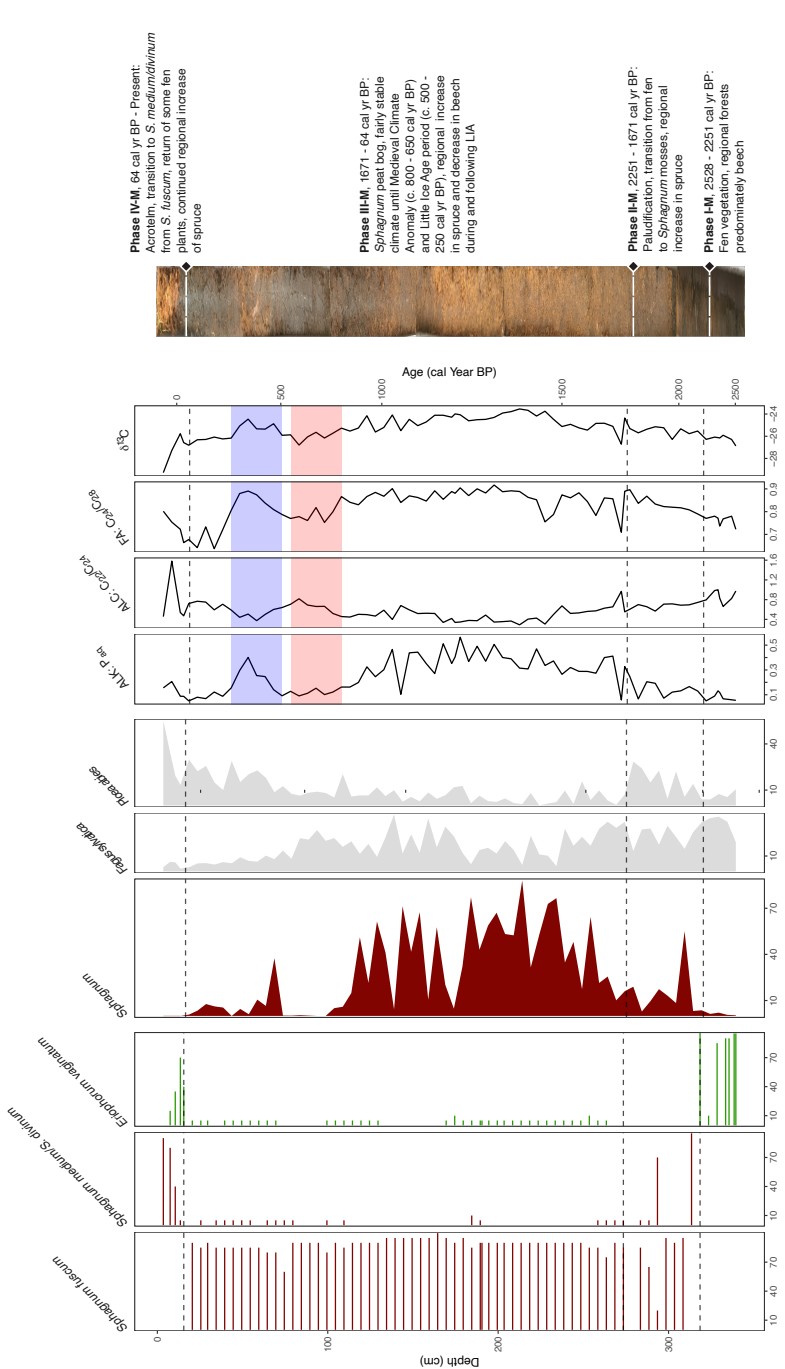

**Figure 8.** Selected proxy curves from the macrofossil record (horizontal bars: *S.fuscum, S. medium/divinum, E. vaginatum*), Pollen Record (filled in lines: *Sphagnum, F. sylvatica, P. abies*), and the Biomarker and elemental records (lines; *n*-alkane $P_{aq}$, *n*-alcohol $C_{22}/C_{24}$, *n*-fatty acid $C_{24}/C_{28}$, and $\delta^{13}C$). A composite photo of the core is also shown with the phases from the CONISS analysis of the macrofossils briefly described. The estimated time frames of the MCA and LIA highlighted on the biomarker and elemental curves in red and blue, respectively.

Additionally, all of the ACL measures reach a minimum at this same interval (Fig. 6), indicating a higher input of the shorter chain length homologues, which would also correspond with less input from vascular plants. Furthermore, the $C_{23}/C_{25}$ ratio reached a minimum in this interval, indicating a potential shift in *Sphagnum* species (McClymont et al., 2008). Following this brief increase, the $P_{aq}$ decreased again while the $P_{wax}$ increased until Phase IV-B or 35 cal yr BP (Fig. 7). This is likely linked to the increased human activity and drainage that occurred at the peatland in the 19th and 20th centuries (Swindles et al., 2019). The indication of the $P_{aq}$, $P_{wax}$, and $C_{23}/(C_{27}+C_{31})$ of a change in local conditions that are not reflected in the macrofossils shows that biomarkers can be a valuable additional proxy to use in paleoenvironmental studies.

## 4.4  Comparison to other local and regional records

The developmental trajectory of the Beerberg peatland was similar to other peat bogs in Germany and other parts of central and northern Europe. The fen to bog transition occurred around 2250 cal yr BP, which has been recognized as a the beginning of a shift to cooler and/or wetter periods in previous studies in Ireland (Barber et al., 2003), England (Wimble, 1986), and Denmark (Aaby, 1976). The next major transition seen in the core record was the Medieval Climate Anomaly (MCA) and Little Ice Age (LIA), the exact time of which varies across European paleoclimate records (Wanner et al., 2022). Here at the Beerberg peatland, we found evidence of warming in the biomarker measurements and the pollen record during the period c. 1150–1300 AD, as shown by the decrease in *Sphagnum* in the pollen record as well as corresponding changes in the biomarker ratios. This was followed by a period of likely cooling from c. 1450–1700 AD, shown by the return of *Sphagnum* in the pollen record and corresponding fluctuations in the biomarker ratios. We believe that these periods correspond to the MCA and LIA. This roughly overlaps with results from previous studies: for example, Mauquoy et al. (2002) identified the LIA as lasting from c. 1450-1600 at Lille Vildmose in Denmark and c. 1460-1600 at Walton Moss in England, Barber et al. (2004) identified the LIA as starting between c. 1250–1350 at Dosenmoor in Germany and Svanemose in Denmark, and De Vleeschouwer et al. (2009) found evidence of the LIA from c. 1200–1800. Following the LIA, the Beerberg pollen record showed that the forests of the region transitioned from primarily *F. sylvatica* forests to *P. abies* forests, with an increase in *P. sylvestris* in the last centuries as well. This pattern has been seen in many other northern and central European forests (e.g., Svobodová et al., 2002). The final vegetation transition at Beerberg beginning in the late 19th century, with an increase in Ericaceae species and *E. vaginatum* as well as a shift in dominant *Sphagnum* species, is likely related at least partially to human influence such as drainage of the peatlands and change in forest composition in the catchment. This has also been noted in other European records (Barber et al., 2004). Generally, the development of the Beerberg peatland confirmed patterns seen at other European sites, though locally transitions occurred with some offsets because of the exposed location of the sequence at the mountain ridge. As such offsets have been well documented, this shows that studying local archives are most informative of local climate and environmental shifts. To the best of our knowledge, this study is the first of its kind covering the last two millennia in a peatland in Central Germany, and its location in Thuringia enables future application to research identifying climate boundary shifts in Central Europe (Breitenbach et al., 2019).

 **5 Conclusions**

We found that the peatland itself, with *Sphagnum fuscum* as the dominant peat-forming species, did not undergo much vegetational change following its initial development. This stability persisted even amidst notable shifts in forest composition, from being beech- to spruce-dominated, and increased anthropogenic land use within the region. In the last couple of centuries, the pristine plant population of the peatland was disturbed, most likely by dust deposition and hydrological changes, as we were able to glean from the elemental and biomarker analyses despite the relative homogeneity of the macrofossil analysis.

This study has further demonstrated opportunities for biomarker analysis to contribute meaningfully to paleoenvironmental investigations. Specifically, the biomarker record serves as an independent confirmation of the trends found in the pollen and macrofossils, providing more confidence in the vegetation reconstruction, especially to species level. The *n*-alkane ratios provided more precise information about fluctuations in local conditions of the peat bog, pointing to potential influences from regional climate shifts that underpin the observed changes in vegetation from the pollen and macrofossil data. We also identified two more potential diagnostic biomarker ratios for *Sphagnum* abundance: *n*-alkanol $C_{22}/C_{24}$ and *n*-fatty acid $C_{24}/C_{28}$. Additionally, the fact that multiple parameters, such as various sources of organic matter and processes like degradation and preservation of organic matter, can be assessed confirms the high potential for biomarker applications in peat records. However, the numerous molecular proxies derived from biomarker composition are often difficult to interpret independently, requiring certain expertise and the assessment of biomarkers from several compound classes to gain supportive data for certain interpretations.

Consequently, to increase the effectiveness and efficiency of biomarker analyses, a more systematic approach is required, aiming at integrating biomarker compound classes more holistically. The vast majority of biomarker indices and ratios have been developed for *n*-alkanes, neglecting other compounds such as *n*-fatty acids and *n*-alkanols. While the *n*-alkanes were the most useful compound class for our study, it would be beneficial if other compounds and potential diagnostic ratios were investigated and applied more systematically, as we have done here.

*Data availability.* All data is available at Pangaea via https://doi.org/10.1594/PANGAEA.961142.

**Appendix A: Supplementary data**

**A1 Radiocarbon dates**

**Table A1.** Radiocarbon dates of selected organic remains. The sample from depth 7.5 cm was excluded from the age-depth model.

| Depth (cm) | Sample ID | Dated material | Radiocarbon date | Error |
|---|---|---|---|---|
| 7.5 | 6845 | *Sphagnum* stems, *Polytrichum* stems | -552 | 23 |
| 16.5 | 6846 | *Sphagnum* stems | 105 | 23 |
| 34.5 | 6847 | *Sphagnum* stems | 173 | 23 |
| 54.5 | 6848 | *Sphagnum* stems | 329 | 23 |
| 69.5 | 6849 | *Sphagnum* stems | 445 | 23 |
| 124.5 | 6850 | *Sphagnum* stems | 1107 | 23 |
| 174.5 | 6851 | *Sphagnum* stems | 1275 | 23 |
| 258.5 | 6852 | *Sphagnum* stems | 1739 | 24 |
| 278.5 | 6853 | *Sphagnum* stems | 1723 | 24 |
| 293.5 | 6854 | *Sphagnum* stems | 1889 | 24 |
| 314.5 | 6855 | *Sphagnum* stems, *Pleurozium schreberi* stems | 2197 | 24 |
| 334 - 336 | 6856 | Charcoal pieces, *Eriophorum vaginatum* spines | 2473 | 24 |

*Author contributions.* **Carrie L. Thomas:** Conceptualization, Formal analysis, Investigation, Data curation, Writing–Original draft, Writing–Review & Editing, Visualization; **Boris Jansen:** Writing–Review & Editing, Supervision; **Sambor Czerwiński:** Formal analysis, Investigation, Writing–Review & Editing, Visualization; **Mariusz Gałka:** Methodology, Formal analysis, Investigation, Writing–Review & Editing, Visualization; **Klaus-Holger Knorr:** Conceptualization, Methodology, Writing–Review & Editing; **E. Emiel van Loon:** Writing–Review & Editing, Supervision; **Markus Egli:** Formal analysis, Investigation, Writing–Review & Editing; **Guido L. B. Wiesenberg:** Conceptualization, Methodology, Resources, Writing–Review & Editing, Supervision, Project administration, Funding acquisition.

*Competing interests.* The authors declare that they have no conflict of interest.

*Acknowledgements.* We are grateful for funding from the Swiss National Science Foundation for the project entitled "IQ-SASS - Improved Quantitative Source Assessment of organic matter in Soils and Sediments using molecular markers and inverse modeling" under contract 188684 and from swissuniversities in the form of a grant supporting the cotutelle de thèse project of CLT. We thank the Vessertal-Thuringian Forest Biosphere Reserve for allowing us to sample, as well as Dr. Marcin Kiedrzyński for his assistance during fieldwork. We are grateful to Thomy Keller for his assistance in obtaining radiocarbon dates. Additional laboratory support was provided by Yves Brügger, Aline Hobi, Tatjana Kraut, Barbara Siegfried, and Dr. Dmitry Tikhomirov. CLT thanks Tiia Määttä for her helpful comments and peatland insight.

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
