# Peer review of "Comparison of paleobotanical and biomarker records of mountain peatland and forest ecosystem dynamics over the last 2600 years in Central Germany"

_Biogeosciences, 2023_

## Author Response (AR1)

"I really liked reading this manuscript. The text is clear and nicely structured, methods are presented in sufficient detail, while the idea behind this study is definitely worth exploring. I have two main observations though. The first is that, in my opinion, the manuscript needs a stronger justification for its novelty. The second observation concerns the biomarker dataset which, despite being one of the fewest of such kind in the temperate Europe, is discussed somewhat superficially. I would have liked to see an extension of the discussion towards exploring potential sources of these biomarker compounds in the study area, backed by some statistics, as pollen and macrofossil data are already available in sufficient resolution."

**Author Response: We have investigated the *n*-alkanols and *n*-fatty acids more thoroughly, including checking for potential diagnostic ratios that are included in the results section and in Fig. 7 on pg. 19 and have expanded the discussion and introduction accordingly.**

Some specific comments below:

Abstract

L. 14-16: I didn't see any evidence in the proxy record for dust deposition. Perhaps consider rephrasing?

**AR: The evidence for dust deposition is the transition in *Sphagnum* species from *S. fuscum* to *S. medium/divinum* in the plant macrofossil data as found by Galka et al., 2019, 2022. As there is not direct evidence, we removed this point from the abstract but left it in the discussion where we explain this more in detail at line 446.**

Methods

2.1. Study area

It would have been nice to see a map with the study area, given that this is a mountainous region and local topography is important in the interpretation of the proxy signals.

**AR: We have added a map as Figure 1 on pg. 5.**

Sampling: It was not explained how the overlapping peat sequences were correlated. Based on stratigraphy?

**AR: The overlapping sequences were correlated based on sampling depth and stratigraphy as they were collected at a distance less than 20 cm from each other. We have added this detail to the methods section at lines 96-97.**

Furthermore, what is the reason behind reporting the averages of geochemistry and biomarker data for the overlapping sequences?

**AR: We averaged the overlapping sequences as there were no clear outliers and to make it easier to treat the data as one core. Because the cores were so close together, we viewed the individual samples from the overlapping sections as replicates. We have also added this to the methods section at lines 97-98.**

2.2 Elemental analysis: please add the sample volume for the samples used for elemental geochemistry.

**AR: The sample mass of 1 mg used for the EA-IRMS analysis has been added at line 103.**

L. 99-100. You mentioned that there are some lithological transitions. I would suggest to add a lithological column to one of the figures (e.g., the geochemistry figure, or the agedepth model).

**AR: We added a composite picture of the core to Fig. 3 on pg 12 and Fig 8 on pg 23.**

L. 102. Consider highlighting more in the introduction the use/purpose of analysing the stable isotopes of N and C. The same suggestion for n-alkanols and n-fatty acids.

**AR: While d15N was measured simultaneously with d13C and the C and N concentrations, the sample weights measured were too small to provide accurate d15N readings and we decided not to include these in the manuscript but neglected to remove d15N from the methods section. We have removed the references to d15N, and added to the introduction.**

2.7.3 Radiocarbon dating: What is the reason for exclusion of the uppermost radiocarbon date?

**AR:  The reason for excluding the uppermost date was that it was from a sample within 10 cm of the surface and yielded a date of -552 +/- 23 cal yr BP, which is not only stratigraphically inconsistent with the rest of the core but is also way too young. The measure is due to the organic matter being formed and incorporated following nuclear weapons testing in the 1950s and 1960s which created a peak of atmospheric 14C. We have explained this in more detail in lines 181- 183.**

Results

Suggestion: When describing the results, focus on time, not on zones. In my opinion, it is more interesting for the reader to know when something happened, rather than the depth intervals.

**AR: We have emphasized the time rather than the depth intervals in the revisions.**

Figure 2. d15N is missing from the figure, although it was mentioned as a performed analysis in the Methods.

**AR: See above comment in reference to L 102.**

Figure 3. It would be useful to see the phases in the macrofossil record drawn on the figure.

**AR: This has been added and is now Figure 4 on page 13.**

Discussion

L. 325-329. 'The disappearance of Neurosporaand Gelasinospora together with the rapid decline in CHAR-micro […] This may also indicate drier conditions on peatlands.' This interpretation is confusing.

**AR: Thank you for pointing this out, it should be wetter and not drier. This has been fixed and is now line 401.**

L.402-403.  $P_{aq}$ and $P_{wax}$ basically mirror each other. Is this situation site-specific? If not, why are both indices necessary?

**AR: This is not site-specific, but they are conventionally both reported in similar studies using biomarkers (e.g., Andersson et al., 2011; Ronkainen et al., 2015; Baker et al., 2016).**

Also, there is no discussion around the *n*-alkanols and *n*-fatty acids.

**AR: As mentioned above, we have further investigated potential diagnostic ratios from the *n*-alkanols and *n*-fatty acids and this has been addressed in the results and discussion.**

References:

Andersson, R. A., & Meyers, P. A. (2012). Effect of climate change on delivery and degradation of lipid biomarkers in a Holocene peat sequence in the Eastern European Russian Arctic. Organic Geochemistry, 53, 63–72.

Baker, A., Routh, J., & Roychoudhury, A. N. (2016). Biomarker records of palaeoenvironmental variations in subtropical Southern Africa since the late Pleistocene: Evidences from a coastal peatland. Palaeogeography, Palaeoclimatology, Palaeoecology, 451, 1–12.

Gałka, M., Szal, M., Broder, T., Loisel, J., and Knorr, K.-H. (2019). Peatbog resilience to pollution and  climate change over the past 2700 years in the Harz Mountains, Germany, Ecological Indicators, 97, 183–193.

Gałka, M., Diaconu, A.-C., Feurdean, A., Loisel, J., Teickner, H., Broder, T., and Knorr, K.-H. (2022). Relations of fire, palaeohydrology, vegetation succession, and carbon accumulation, as reconstructed from a mountain bog in the Harz Mountains (Germany) during the last 6200 years, Geoderma, 424.

"The manuscript reports a multiproxy caracterisation of a peat core from central Germany. Carbon, nitrogen, macrofossil, pollen and biomarker contents along the 320 cm core are interpreted in term of palaeovegetation and palaeoenvironment. Data undoubtly deserve publication and their palaeoecology implications are relevant toward a better understanding of recent vegetation changes under global change. However, the manuscript requires some rewriting before being publishable in Biogeosciences.

While the multiproxy approach is certainly more powerful than a single method approach, it is not the first time it is adopted and claiming it as a main objective and discussion point of the paper appears slightly over the top "

**AR: We have further investigated the n-alkanols and n-fatty acids and have expanded the results and discussion accordingly.**

I suggest to directly integrate biomarkers in the paleoreconstruction of each phase.

**AR: We have rewritten the discussion accordingly.**

The methodology of data processing and depth clustering should be homogenized between proxies, or at least better argued/justified and discussed. Why pollen and biomarker data are clustered through a CONISS approach while depth grouping based

on macrofossil is achieved visually and EA-IRMS were submitted to no clustering approach ?

**AR: We were able to apply the CONISS approach to both the macrofossil and EA data and have updated the methods, figures, results, and discussion accordingly.**

Why is the main phase determination based on macrofossils?

**AR: We based the main phases on the macrofossils because we believed they would be the most reliable proxy reflecting in situ vegetation and peat development as is conventional in paleoecology (e.g. Birks & Birks, 2000).**

The cluster-differences between proxies should be discussed at the beginning of the discussion so as to further directly integrate all proxies in the paleoreconstructions.

**AR: We have started the discussion with a section on the temporal differences between proxies from lines 365-385.**

In addition to the palaeoecological reconstruction phase by phase, peat and vegetation dynamics should be further discussed in relation to other local peats and regional changes. Can general trends be drawn from the studied core (and other previously studied peats) in term of peat evolution under climate change?

**AR: We have added a discussion section comparing the peatland development to other records from lines 526-540.**

Provide sketch palaeoenvironmental reconstruction of each phase

**AR: This has been added to the revised figure 8 on pg. 23.**

Replace "arboreal" by "tree" or "arborescent"

**AR: We respectfully disagree about this as "arboreal" is the conventionally used term in similar paleovegetation reconstructions. Therefore, we have kept the respective terminology.**

Briefly mention biomarker absolute amount before describing their distribution in 3.5.

**AR: This has been added into the results section 3.5.**

Enlarge captions of figure 3 and 4

**AR: We have enlarged both captions and the figures are now numbered 4 and 5 on pgs. 13 and 15, respectively.**

L230: write "C/N" instead of "N" ?

**AR: This has been corrected and is now L. 247**

L358: delete "and"

**AR: This has been corrected and is now L. 445.**

L369: insert "1" before "657"

**AR: This section was removed.**

References

Birks, H. H., & Birks, H. J. B. (2000). Future Uses of Pollen Analysis Must Include Plant Macrofossils. Journal of Biogeography, 27(1), 31–35.

---

## Author Response (AR2)

**Reviewer comment**: I still believe that the manuscript's novelty and the targeted knowledge gaps could be better addressed in the introduction. At the moment, the introduction feels a bit general, and could benefit from highlighting the specific novel aspects that will be gained, or gaps in understanding that will be addressed by this particular multi-proxy study. For example, why is it so important to reconstruct environmental dynamics locally? Are there specific/critical problems that the study area confronts with, which could be better understood as a result of this study? Why the lack of continuous palaeorecords in the study area is a drawback? To what extent could the new information gained from this local study be relevant for peatlands in general?

**Author response: We added justification for the importance of high-resolution paleorecords in the Thuringia region.**

**RC**: In the second paragraph of the introduction, we learn that 'peatlands are expected to be vulnerable as the climate currently changes' and that 'it is crucial to investigate the past vegetation dynamics and peat accumulation in response to past environmental drivers to better understand how they will respond in the future and how this will affect their carbon sink function'. Yet, I feel that, by the end of the manuscript, the link to these general ideas is not clearly shown.

**AR: We have removed this sentence as the general idea is indeed not truly linked to the aims of the paper.**

**RC**: In Fig. 1, I assume the peatland area is delimited by the continuous white line (please add this in the caption). Then, what do the blue horizontal line symbols represent?

**AR: The blue lines have been removed and the caption has been adjusted.**

**RC**: Lines 215-236. Add parentheses in the formulas, to indicate the order of operations. For example, it is (C22+C24)/(C26+C28), and not C22+C24/C26+C28, and many others.

**AR: We have added parentheses where necessary to the formulas.**

**RC**: Section 3.2. For consistency, I would recommend presenting the results for the elemental analysis based on zones/phases, like in the case of the other proxies.

**AR: We have rewritten this section in the style of the others.**

**RC**: Section 4.1. It would be great to actually see the how the statistically identified phases among all the proxies compare. Right now, this section is a bit hard to follow. Perhaps an idea would be adding a table on the age/depth scale, with all the proxy categories, proxy phases and major trends in each proxy category for each phase. Or mark phase boundaries for different proxies in the synthesis figure (Fig. 8).

**AR: Thank you for the suggestion. We have added Table 1 to visualize the differences between the phases and make the section easier to follow.**

**RC**: Lines 401-402, 411, 433-435 are results, and should be discussed.

**AR: We have added discussion for these results.**

**RC**: Lines 447-448, 452, 486 – the same idea of recent peatland drying across Europe is repeated three

times. Consider rephrasing.

**AR: Thank you for pointing out the repetition. We have adjusted the text.**

**RC**: Lines 495, 496, 503– remind the reader what is the proxy evidence that supports these statements.

**AR: We have added the corresponding proxy evidence to these sentences.**